

# Shallow trimers of two identical fermions and one particle in resonant regimes

Pascal Naidon[1][*], Ludovic Pricoupenko[2][†] and Christiane Schmickler[1][‡]

**1** RIKEN Nishina Centre, Strangeness Nuclear Physics Laboratory,
RIKEN, Wakō, 351-0198 Japan
**2** Sorbonne Université, CNRS, Laboratoire de Physique Théorique de la Matière Condensée
(LPTMC), F-75005 Paris, France

[*] pascal@riken.jp , [†] ludovic.pricoupenko@sorbonne-universite.fr , [‡] christiane.schmickler@riken.jp

## Abstract

We consider two identical fermions interacting in the p-wave channel. Each fermion also interacts with another particle in the vicinity of an s-wave resonance. We find that in addition to the Kartavtsev-Malykh universal trimer states resulting from the s-wave particle-fermion interaction, the fermion-fermion p-wave interaction induces one or two shallow trimers in a large domain of the control parameters, including a borromean regime where the ground-state trimer exists in the absence of dimer at any mass ratio between the fermions and the particle. A generic picture of the trimer spectrum emerges from this work in terms of the low-energy parameters of the interactions.



# 1  Introduction

The experimental achievement of Fano-Feshbach resonances in ultra-cold atomic systems [1] have enabled researchers to modify interatomic interactions at will. In particular, it is possible to tune the scattering length of the atoms to values much larger than the interaction range and thus to achieve quantum systems of particles with short-range resonant interactions, whose properties are universally described by only a few parameters. At the two-body level, two atoms undergoing such a resonant interaction in the s-wave can form halo dimers, whose binding energy scales universally with the square inverse of the scattering length. At the three-body level, it is possible to form Efimov trimer states [2], which are universally described by the scattering length and a three-body parameter [3–5]. The most striking property of these states is that they exhibit the Efimov effect: at resonance, where a halo dimer becomes bound, the trimer spectrum forms an infinite geometric series with an accumulation point at vanishing energy. This effect results from the Efimov attraction, an effective three-body attraction due to the resonant nature of the interaction and which arises far outside the range of inter-particle interactions. The experimental confirmation of Efimov trimers [6–27] with ultracold atoms has spurred an intense theoretical search for universal few-body states in systems with resonant interactions.

The Efimov effect is most easily evidenced in bosonic systems, and tends to be absent in fermionic systems, which involve non-zero angular momenta due to the Pauli exclusion. Indeed, a non-zero angular momentum implies a centrifugal repulsion, which often overcomes the Efimov attraction. For instance, three-body systems of spin-1/2 fermions do not exhibit the Efimov effect for this reason. The relative strength of the Efimov attraction with respect to the centrifugal repulsion can nonetheless be enhanced in the case of particles of different masses. The simplest system consists of two identical fermions of mass $M$ in the same spin state interacting with a third particle (fermion or boson) of mass $m$. The third particle experiences an s-wave scattering resonance with each fermion. Analogously to the exchange of mesons between two nucleons in nuclear physics, it mediates an effective attraction between the two fermions. The fact that the two identical fermions are exactly in the same internal state implies that they have at least one unit of relative angular momentum, resulting in a centrifugal repulsion competing with the Efimov attraction. When the mass ratio $x = M/m$ exceeds a critical value $x_c = 13.607\ldots$, the Efimov attraction dominates over the centrifugal repulsion

and the Efimov effect occurs at resonance, resulting in an infinite number of trimers with the $J^\pi = 1^-$ symmetry. At finite but large and positive scattering length, when the mass ratio is smaller than this critical value, the effective three-body attraction persists at intermediate distances, making it possible for the three particles to bind into a finite number of bound states.

Considering two fermions interacting with the third particle only through a contact interaction, Kartavtsev and Malykh have demonstrated that for mass ratios larger than $x_1 = 8.17260...$ and smaller than $x_c$, up to three trimer states can exist [28–31]. These trimers have the same $J^\pi = 1^-$ symmetry as Efimov trimers at larger mass ratios and are characterised by the scattering length $a$ between each of the two fermions and the third particle, and additionally, for $M/m$ larger than $x_r = 8.619\ldots$, by a three-body parameter. These universal results are applicable to describe shallow trimer states in real systems in the limit of large scattering length $a$ and negligible interactions between the fermions. There have been some attempts [32–34] to understand more precisely how this universal scenario fits into real systems. In what follows, we will call *s-wave induced trimers* the states bound only by the s-wave interaction between the fermions and particle, and use the short-hand notation *KM states* or *KM limit* to refer to their universal limit described by Kartavtsev and Malykh's contact theory.

One pending question is how the universal scenario is modified by the presence of an interaction between the two fermions. In real systems, an interaction is indeed always present. If sufficiently attractive, in the vicinity of a two-body p-wave scattering resonance, this interaction may even bind the two fermions, despite their centrifugal repulsion. This attractive effect is described at low energy by only two parameters, the p-wave scattering volume and the p-wave effective range. One may thus wonder whether the universality of the KM states is preserved in the presence of the fermions' interaction, i.e. whether the system can still be universally described by a few low-energy parameters.

In this paper, we show that the universal KM states always exist for the mass ratios predicted by the contact theory and for a sufficiently large and positive value of the scattering length. However, for sufficiently attractive p-wave interaction, the spectrum is enriched by another shallow state whatever the mass ratio, and by a second one for a mass ratio $M/m$ larger than a critical value $x_c'$. Hereafter, we will use the denomination *p-wave induced trimers* for these states. Their properties (threshold, energy...) depend on the shape of the p-wave interaction. Despite this non-universality, these results permit us to draw a generic picture for the spectrum of the shallow trimers in terms of s-wave and p-wave induced trimers, as a function of the two-body low-energy parameters. Both the s-wave and p-wave induced trimers are depicted schematically in Fig. 1 as a function of the s-wave scattering length $a$ and the p-wave scattering volume $v$. As will be discussed, these two types of trimers may undergo an avoided crossing and hybridise when their energies come close.

## 2 Model

### 2.1 Separable model

We consider the general problem of two identical fermions of mass $M$ and one particle of mass $m$, all three interacting with each other through finite-range attractive interactions. The two fermions are denoted as particles 1 and 2, and the third particle as particle 3. In these notations, the Schrödinger equation at energy $E$ reads in momentum representation:

$$\left( \frac{\hat{p}_1^2}{2M} + \frac{\hat{p}_2^2}{2M} + \frac{\hat{p}_3^2}{2m} + \hat{U}_{12} + \hat{V}_{23} + \hat{V}_{31} - E \right) |\Psi\rangle = 0, \tag{1}$$

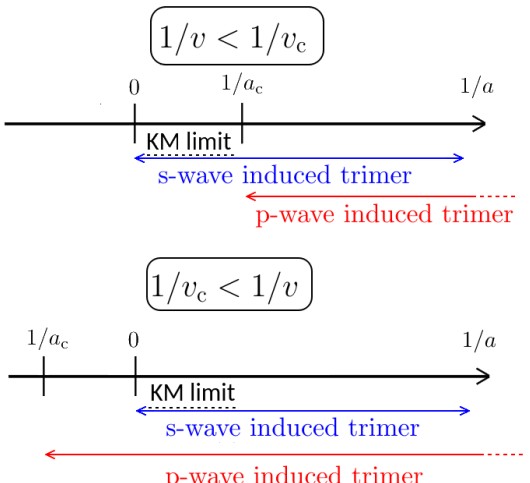

Figure 1: Domains of existence of the two types of trimers identified in this work, for a mass ratio $M/m \in [x_1 \approx 8.1726, x'_c < x_2]$. The *s-wave induced trimer*, shown schematically in blue, is induced by the s-wave interaction between the fermions and the particle. It exists only for positive s-wave scattering lengths $a > 0$ and admits a universal limit (KM limit) for $1/a \to 0^+$. The *p-wave induced trimer*, shown in red, is induced by the p-wave interaction between the fermions in presence of the s-wave interaction. It exists only for positive $1/a > 1/a_c$ when the p-wave inverse scattering volume $1/v$ is less than a critical value $1/v_c$ (upper panel), and can exist for negative $1/a$ (Borromean trimer) when $1/v > 1/v_c$ (lower panel). Not shown in the figure: for larger mass ratios $M/m > x_2$, there is a second s-wave induced trimer, converging to the excited KM state for $1/a \to 0^+$. For mass ratios $M/m > x'_c$, there is an excited p-wave induced trimer. Finally, for $M/m > x_c \approx 13.607$, there are no more KM states for $1/a \to 0^+$ but an infinite number of s-wave induced trimers converging to Efimov states.

where $\hat{p}_i$ is the momentum operator of particle $i$, and $\hat{V}_{ij}$ and $\hat{U}_{ij}$ are the interaction operators between particles $i$ and $j$. No three-body interaction is considered in this work. The state $|\Psi\rangle$ is antisymmetric under the exchange of fermions, i.e. $\langle k_2, k_1, k_3 | \Psi \rangle = -\langle k_1, k_2, k_3 | \Psi \rangle$ in momentum representation where $k_i$ is the wave vector of particle $i$. Because of translational invariance, we can pursue this study in the center-of-mass frame and thus $k_1 + k_2 + k_3 = 0$. The wave function can then be expressed in terms of one of the three Jacobi coordinate sets, $(k_i, k_{jk})$, where $k_{jk} \equiv \frac{m_j k_k - m_k k_j}{m_j + m_k}$ is the relative momentum between particles $j$ and $k$, with $m_1 = m_2 = M$ and $m_3 = m$.

To simplify the calculations, the interaction operators are taken to be of the separable type:

$$\hat{V} = \frac{2\pi\hbar^2}{\mu_{23}} \xi |\chi\rangle\langle\chi|, \tag{2}$$

$$\hat{U} = \frac{6\pi\hbar^2}{\mu_{12}} \sum_{m=-1}^{1} g_m |\Phi_m\rangle\langle\Phi_m|, \tag{3}$$

where $\mu_{23} = \left(\frac{1}{M} + \frac{1}{m}\right)^{-1}$ and $\mu_{12} = M/2$ are the reduced masses of the fermion-particle and two-fermion subsystems. The separable potentials are characterised by interaction strengths $\xi \leq 0$ and $g_m \leq 0$, and form factors $\chi$ and $\Phi_m$, with $\langle k|\chi\rangle \equiv \chi(k)$ and $\langle k|\Phi_m\rangle \equiv \Phi_m(k) = \phi_m(k) k \cdot \hat{e}_m$, where we use the unit vectors $\hat{e}_0 = \hat{e}_z$ and

$\hat{e}_{\pm 1} = (\mp \hat{e}_x - i\hat{e}_y)/\sqrt{2}$. The form of these separable potentials is chosen such that $\hat{V}$ only affects the s wave, and $\hat{U}$ only affects the p wave - see Appendix 1. In general the interaction $\hat{U}$ can be anisotropic, and becomes isotropic in the case where all the interaction strengths $g_m$ and form factors $\phi_m$ are equal. In this paper, without loss of generality we will choose the normalisation of the form factors such that $\phi_m(0) = \chi(0) = 1$.

Thanks to the separability of the potentials, the problem can be formulated in terms of the quantities,

$$S(\mathbf{k}_1) = -\xi \int \frac{d^3 k_{23}}{(2\pi)^3} \langle \chi | \mathbf{k}_{23} \rangle \langle \{\mathbf{k}_i\} | \Psi \rangle , \tag{4}$$

$$P_m(\mathbf{k}_3) = -g_m \int \frac{d^3 k_{12}}{(2\pi)^3} \langle \Phi_m | \mathbf{k}_{12} \rangle \langle \{\mathbf{k}_i\} | \Psi \rangle . \tag{5}$$

Physically, $S$ corresponds to the relative wave function between a fermion-particle pair in a small volume and the other fermion, and $P_m$ to the relative wave function between the two-fermion pair in a small volume and the third particle when the fermion pair has an angular momentum projection $m\hbar$. They satisfy the generalised Skorniakov–Ter-Martirosian (STM) equations (see appendix 2):

$$-\frac{|\chi(i\kappa_0)|^2}{4\pi f_{0,0}(i\kappa_0)} S(\mathbf{k}) = y \int \frac{d^3 k'}{(2\pi)^3} \frac{\langle \chi | \mathbf{k}' + \frac{x}{2y}\mathbf{k} \rangle \langle \mathbf{k} + \frac{x}{2y}\mathbf{k}' | \chi \rangle S(\mathbf{k}')}{y(k^2 + k'^2) + x\mathbf{k} \cdot \mathbf{k}' + q^2} \tag{6}$$

$$+ 3 \int \frac{d^3 k'}{(2\pi)^3} \frac{\langle \chi | \mathbf{k}' + \frac{1}{2y}\mathbf{k} \rangle \sum_m \langle \mathbf{k} + \frac{1}{2}\mathbf{k}' | \Phi_m \rangle P_m(\mathbf{k}')}{k^2 + yk'^2 + \mathbf{k} \cdot \mathbf{k}' + q^2} ,$$

$$\frac{|\kappa_1 \phi_m(i\kappa_1)|^2}{4\pi f_{1,m}(i\kappa_1)} P_m(\mathbf{k}) = 2y \int \frac{d^3 k'}{(2\pi)^3} \frac{\langle \Phi_m | \mathbf{k}' + \frac{\mathbf{k}}{2} \rangle \langle \mathbf{k} + \frac{\mathbf{k}'}{2y} | \chi \rangle S(\mathbf{k}')}{yk^2 + k'^2 + \mathbf{k} \cdot \mathbf{k}' + q^2} , \tag{7}$$

where we have introduced the binding wave number $q$ of the system defined by

$$E = -\frac{\hbar^2 q^2}{M} , \tag{8}$$

as well as the following short-hand notations for numerical factors depending only on the mass ratio,

$$x \equiv M/m , \tag{9}$$

$$y \equiv (x+1)/2 . \tag{10}$$

There are two kinds of terms in the STM equations: the terms on the right-hand side, which contain integrals describing the three-body sector, and the terms on the left-hand side, which describe only the two-body sector and contain the s-wave and p-wave two-body scattering amplitudes $f_{0,0}$ and $f_{1,m}$ given by

$$-\frac{|\chi(i\kappa)|^2}{f_{0,0}(i\kappa)} = \frac{1}{\xi} + \frac{2}{\pi} \int_0^\infty dk\, k^2 \frac{|\chi(k)|^2}{k^2 + \kappa^2} , \tag{11}$$

$$\frac{|\kappa \phi_m(i\kappa)|^2}{f_{1,m}(i\kappa)} = \frac{1}{g_m} + \frac{2}{\pi} \int_0^\infty dk\, k^2 \frac{|k\phi_m(k)|^2}{k^2 + \kappa^2} . \tag{12}$$

These amplitudes are evaluated at the relative wave number $i\kappa$ between two particles in the presence of a third particle of wave number $k$ with respect to that pair, for a fixed three-body

energy $E$. This binding wave number is given, for the s-wave and p-wave respectively, by $\kappa_0$ and $\kappa_1$ satisfying

$$\kappa_0^2 \equiv \frac{2x+1}{4y^2}k^2 + q^2/y\,, \tag{13}$$

$$\kappa_1^2 \equiv \frac{2x+1}{4}k^2 + q^2\,. \tag{14}$$

In what follows, we will introduce the scale $\Lambda_0$ (resp. $\Lambda_1$) below which the form factor $\chi(k)$ (resp. $\phi(k)$) are almost constant (i.e. equal to unity in our choice of normalisation). Physically these scales are related to the radius of the actual interactions which are supposed to be of short range. In the context of ultracold atoms they are of the order of the inverse of the van der Waals lengths $\Lambda_0 \propto \left(\frac{\hbar^2}{2\mu_{23}C_{23}}\right)^{1/4}$ and $\Lambda_1 \propto \left(\frac{\hbar^2}{2\mu_{12}}C_{12}\right)^{1/4}$, where $C_{23}$ and $C_{12}$ are the dispersion coefficients of the $-1/r^6$ van der Waals term in the respective pairwise potential for the pairs of particles (12) and (23). In the small momentum limit where $\kappa$ is much smaller than $\Lambda_0$ and $\Lambda_1$, one has the usual expansions of the scattering amplitudes:

$$\frac{1}{f_{0,0}(i\kappa)} = -\frac{1}{a} + \kappa - \frac{r_e}{2}\kappa^2 + o(\kappa^2)\,, \tag{15}$$

$$\frac{-\kappa^2}{f_{1,m}(i\kappa)} = -\frac{1}{v_m} + \alpha_m \kappa^2 - \kappa^3 + o(\kappa^3)\,, \tag{16}$$

where $a$ is the s-wave scattering length and $v_m$ is the scattering volume, which in general depends on the quantum number $m$ of the fermionic pair. Physical short-range potentials admit the same expansions as Eqs. (15-16). At the two-body level, the separable potentials are therefore indistinguishable from physical ones in this low-momentum limit. This motivates their choice for the study of low-energy physics. However, the separability induces discrepancies at higher momenta comparable or larger than $\Lambda_0, \Lambda_1$, which may affect the properties of low-energy three-body states. For instance, the three-body parameter of Efimov states may differ, although these differences are generally small as long as the scale $\Lambda_0^{-1}$ is properly set to the physical range of interactions. Separable s-wave potentials have been shown to be a good approximation for shallow potentials supporting at most one dimer [35, 36]. We assume that in a similar fashion, the p-wave separable potential is an adequate approximation for a qualitative description of low-energy three-body states, as long as the scale $\Lambda_1^{-1}$ corresponds to the physical range of interaction.

By taking the limit $\kappa \to 0$ of Eqs. (11-12), one finds the relations between the interaction strengths $\xi$ and $g_m$ and the scattering length and volume $a$ and $v_m$:

$$\frac{1}{a} = \frac{1}{\xi} + \frac{2}{\pi}\int_0^\infty dk|\chi(k)|^2\,, \tag{17}$$

$$\frac{1}{v_m} = \frac{1}{g_m} + \frac{2}{\pi}\int_0^\infty dk k^2|\phi_m(k)|^2\,. \tag{18}$$

The coefficients $r_e$ and $1/\alpha_m$ in Eqs. (15,16) are the s-wave and p-wave effective ranges, which in general depend on $a$ and $v_m$, respectively. Although the effective ranges are useful to describe the low-energy two-body physics, as their name implies, they do not represent the true range of interactions. For near-resonant three-body systems, it is more useful to consider the value $\bar{r}_e$ of the effective range $r_e$ at the s-wave resonance ($1/a \to 0$), and the value $1/\bar{\alpha}_m$ of the effective range $1/\alpha_m$ at the p-wave resonance ($1/v_m \to 0$). For the single-channel interactions considered in the present study, $\bar{r}_e$ and $1/\bar{\alpha}_m$ are of the order of the potential radii $1/\Lambda_0$ and $1/\Lambda_1$, and have, unlike $\Lambda_0$ and $\Lambda_1$, a precise definition from Eqs. (15,16). For this reason, we

will compare different interaction models having the same scattering lengths/volumes, and the same effective ranges in the limits $1/a \to 0$ or $1/v_m \to 0$. This ensures that these models have the same two-body spectra near these limits and also have the same interaction ranges. In this spirit, the s-wave effective range at resonance $\bar{r}_e$ will be used as the unit of length throughout this paper.

## 2.2 Isotropic p-wave interaction

The isotropic p-wave interaction where $g_m = g$ and $\phi_m = \phi$ plays a central role in the subsequent analysis. In this section, we thus simplify the STM equations by using symmetry considerations in this particular case. Moreover, we will show that the results in the isotropic limit can be directly used for a more realistic anisotropic p-wave interaction by using a perturbative treatment. In what follows, we will use the notation $f_{1,m} \equiv f_1$, $v_m \equiv v$ and $\bar{\alpha}_m = \bar{\alpha}$. Before going to the three-body equations, it is worth recalling some important and general properties of isotropic p-wave interactions in the resonant regime [37–39]:

   $i$) the scattering resonance due to a quasi-bound state occurs for large and negative values of the scattering volume ($1/v \to 0^-$) at a specific value of the relative momentum $k_{\text{res}} = 1/\sqrt{-\bar{\alpha}v}$;

  $ii$) the width of the scattering resonance is of the order of $k_{\text{res}}/\bar{\alpha}$, which is why the inverse p-wave effective range at resonance $\bar{\alpha}$ [see Eq. (16)] may also be called shortly the *width parameter*;

 $iii$) for a short range potential of radius $R$ (in this paper, $R$ is of the order of $1/\Lambda_1$), the width parameter verifies the 'width-radius inequality' $\bar{\alpha}R \gtrsim 1$ (the inequality is not strict, depending on the precise definition chosen for the radius $R$). This inequality was obtained by using the modified scalar product introduced in the contact potential approach [38] and it also corresponds to the Wigner bound [40, 41] imposed by the positivity of the probability density. For the model potentials used in this study, $\bar{\alpha}$ is of the order of $\Lambda_1$. In more general situations, for instance for a multichannel interaction, this parameter can be much larger than $\Lambda_1$, corresponding to a narrow resonance limit;

  $iv$) In the limit ($1/v \to 0^+$), there is no p-wave scattering resonance and there is a shallow p-wave dimer of binding wave number $1/\sqrt{\bar{\alpha}v}$.

Hence, the resonant regime in the p-wave scattering differs from the one in the s wave where the unitary limit can be reached in a large range of the momentum and not only for a specific value. In what follows, due to the continuity found in the trimer spectrum at $1/v = 0$ (thus including arbitrarily large and negative or positive scattering volumes), we will formally qualify this limit as the *p-wave resonance limit*.

We now turn to the simplification of the STM equation for an isotropic p-wave interaction. For convenience, we introduce the spinor

$$P(k) = \sum_{m=-1}^{1} P_m(k)\hat{e}_m \, . \tag{19}$$

The equations (6-7) thus become equations on $S(k)$ and $P(k)$:

$$-\frac{|\chi(i\kappa_0)|^2}{4\pi f_{0,0}(i\kappa_0)} S(k) = y \int \frac{d^3 k'}{(2\pi)^3} \frac{\chi^*(|k' + \frac{x}{2y}k|)\chi(|k + \frac{x}{2y}k'|)S(k')}{y(k^2 + k'^2) + x k \cdot k' + q^2} \tag{20}$$

$$+ 3 \int \frac{d^3 k'}{(2\pi)^3} \frac{\chi^*(|k' + \frac{1}{2y}k|)\phi(|k + \frac{1}{2}k'|)(k + \frac{1}{2}k') \cdot P(k')}{k^2 + y k'^2 + k \cdot k' + q^2} \, ,$$

$$\frac{|\kappa_1\phi(i\kappa_1)|^2}{4\pi f_1(i\kappa_1)}P(\boldsymbol{k}) = 2y\int\frac{d^3k'}{(2\pi)^3}\frac{\left(\boldsymbol{k}'+\frac{\boldsymbol{k}}{2}\right)\phi^*(|\boldsymbol{k}'+\frac{\boldsymbol{k}}{2}|)\chi(|\boldsymbol{k}+\frac{\boldsymbol{k}'}{2y}|)S(\boldsymbol{k}')}{yk^2+k'^2+\boldsymbol{k}\cdot\boldsymbol{k}'+q^2}, \tag{21}$$

where we used $\Phi_m(\boldsymbol{k}) = \phi(k)\boldsymbol{k}\cdot\hat{\boldsymbol{e}}_m$.

In this work, we consider the states of total angular momentum $J = 1$ and negative parity, consistent with the known symmetry of the KM trimers. Therefore, when the relative angular momentum between one fermion and the third particle is zero, the remaining angular momentum between the fermion-particle pair and the other fermion must be unity. Assuming that the total momentum projection is zero along a fixed unit vector $\hat{\boldsymbol{e}}_z$, $S(\boldsymbol{k})$ must be of the form:

$$S(\boldsymbol{k}) = s(k)\cos\theta, \tag{22}$$

where $\theta$ is the angle between $\boldsymbol{k}$ and the fixed vector $\hat{\boldsymbol{e}}_z$.

Likewise, when the relative angular momentum between the two fermions is unity, the remaining angular momentum between the two-fermion pair and the third particle can be either zero or two. It follows that $P(\boldsymbol{k})$ is of the form:

$$P(\boldsymbol{k}) = p_0(k)\hat{\boldsymbol{e}}_z + p_2(k)[\hat{\boldsymbol{e}}_k\times(\hat{\boldsymbol{e}}_k\times\hat{\boldsymbol{e}}_z)], \tag{23}$$

where $\hat{\boldsymbol{e}}_k = \boldsymbol{k}/k$ - see Appendix 3 for details. Inserting Eqs. (22) and (23) into Eqs. (20-21) yields a set of three integral equations for $s$, $p_0$ and $p_2$:

$$\frac{|\chi(i\kappa_0)|^2}{f_0(i\kappa_0)}s(k)+\int_0^\infty\frac{dk'}{\pi}k'^2\Big[L(k,k')s(k'+\big(3L_0(k,k')-L_2(k,k')\big)p_0(k')$$
$$+\big(L_2(k,k')-2L_0(k,k')\big)p_2(k')\Big]=0, \tag{24}$$

$$3\frac{|\kappa_1\phi(i\kappa_1)|^2}{2yf_1(\kappa_1)}p_0(k)-\int_0^\infty\frac{dk'}{\pi}k'^2L_0(k',k)^*s(k')=0, \tag{25}$$

$$3\frac{|\kappa_1\phi(i\kappa_1)|^2}{yf_1(\kappa_1)}p_2(k)-\int_0^\infty\frac{dk'}{\pi}k'^2L_2(k',k)^*s(k')=0. \tag{26}$$

The kernels $L$, $L_0$, and $L_2$ are given by:

$$L(k,k')=\int_{-1}^1 du\,yu\frac{\chi^*(|\boldsymbol{k}'+\frac{x}{2y}\boldsymbol{k}|)\chi(|\boldsymbol{k}+\frac{x}{2y}\boldsymbol{k}'|)}{y(k^2+k'^2)+xkk'u+q^2}, \tag{27}$$

$$L_0(k,k')=3\int_{-1}^1 du\left(ku^2+\frac{1}{2}k'u\right)\times\frac{\chi^*(|\boldsymbol{k}'+\frac{1}{2y}\boldsymbol{k}|)\phi(|\boldsymbol{k}+\frac{1}{2}\boldsymbol{k}'|)}{k^2+yk'^2+kk'u+q^2}, \tag{28}$$

$$L_2(k,k')=3\int_{-1}^1 du\left(k\big(3u^2-1\big)+k'u\right)\times\frac{\chi^*(|\boldsymbol{k}'+\frac{1}{2y}\boldsymbol{k}|)\phi(|\boldsymbol{k}+\frac{1}{2}\boldsymbol{k}'|)}{k^2+yk'^2+kk'u+q^2}, \tag{29}$$

where $u = \boldsymbol{k}\cdot\boldsymbol{k}'/(kk')$ is the cosine of the angle between $\boldsymbol{k}$ and $\boldsymbol{k}'$.

## 2.3 Form factors

Although the values of the interaction strengths $\xi$ and $g$ can be set to reproduce a given scattering length and scattering volume through Eqs. (17-18), the form factors $\chi$ and $\phi$ remain to be chosen. We consider four different types of form factors, referred to as the "Gaussian", "Yamaguchi", "Yamaguchi-squared", and "Cut-off" models. Their expressions are given in Tables 1 and 2 of Appendix 1. The tables also provide the explicit forms of the

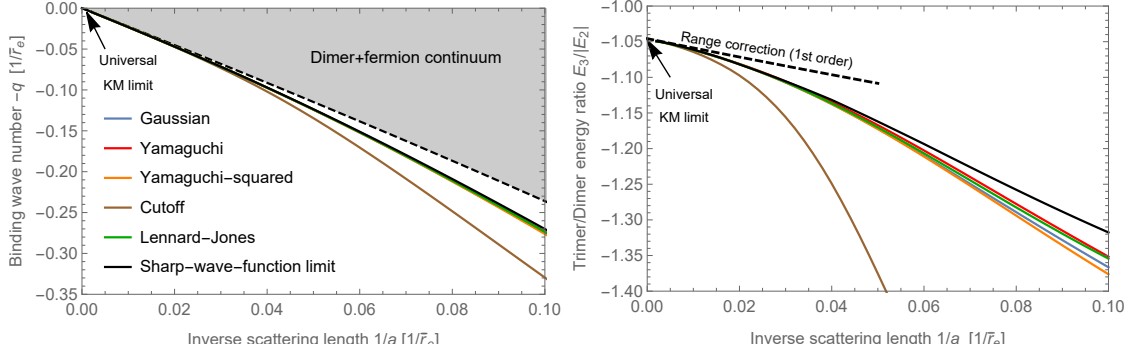

Figure 2: Trimer spectrum without fermion-fermion interaction for a mass ratio $x = 9$. Left panel: trimer's binding wave number $q$ as a function of the inverse of the scattering length $a$ for different models (note that the curves corresponding to some models cannot be distinguished in this panel - see right panel). Dashed curve: threshold of the dimer+fermion scattering continuum (shaded area) obtained from the binding wave number of the fermion-particle s-wave dimer in the effective-range approximation (see main text); Solid curves: trimer binding wave number for different models of the fermion-particle interaction (see Table 1 and main text). Right panel: ratio between the trimer and dimer energies $E_3/|E_2|$ as a function of the inverse scattering length; Dashed curve: first-order correction in $\bar{r}_e/a$ with respect to the zero-range potential limit (see Eqs. (31-32) and main text).

two-body scattering amplitudes $f_0$ and $f_1$, as well as the corresponding parameters $\bar{r}_e$ and $\bar{\alpha}$. In addition to these simple model potentials, a more realistic separable potential is also considered, which is constructed to reproduce exactly the wave function scattered at zero energy by a Lennard-Jones potential. The details of this separable potential have been given in [42].

## 3  Absence of interaction between the fermions

In this section, we consider the regime where the interaction between the fermions can be neglected. Thus we will use Eq. (6) with $P_m = 0$, that is to say, Eq. (24) with $p_0 = p_2 = 0$.

In this regime, the results are expected to conform to the Kartavtsev-Malykh universal theory for large enough scattering lengths $a$. Indeed, in the small momentum and energy limits where $k, k', q, 1/a \ll \Lambda_0$, Eq. (24) is equivalent to the zero-range limit of the STM equation with $\Lambda_0 \to \infty$ for a fixed value of $a$. This can be understood as follows: first, consider the small-energy $q \ll \Lambda_0$ and small-momentum limit $k \ll \Lambda_0$ of Eq. (24). For such small wave numbers, $|\chi|^2/f_0$ may be approximated by $-1/a + \kappa_0$ according to Eq. (15). Although the integral over $k'$ extends to infinity, i.e. values of $k'$ where $s(k')$ is not described by the small-energy and small-momentum approximation, it turns out that for $1/a \ll \Lambda_0$, $s(k)$ is peaked around $k \sim 1/a$ and then decays to zero. The contributions from $k' \gg 1/a$ are thus suppressed by $s(k')$ and since $1/a \ll \Lambda_0$, the form factors $\chi$ inside $L(k, k')$ may be approximated to unity. It follows that the small-momentum approximation of the STM equation is self-consistent and properly describes the limit of the STM equation for small energy and large scattering length.

This approximated equation formally corresponds to the zero-range STM equation obtained for two-body contact interactions and no three-body contact interaction. It is thus equivalent to the Kartavtsev–Malykh contact theory with a zero three-body parameter. From this, we conclude that the finite range of the s-wave interactions does not lead to a non-zero three-body

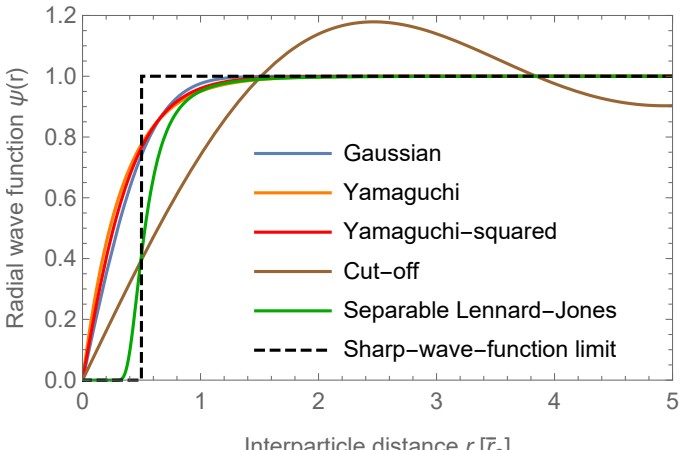

Figure 3: S-wave radial two-body wave function at zero energy obtained for different models at the s-wave resonance ($|a| \to \infty$) - see Table 1 and main text. The wave function is normalised to tend to unity asymptotically, and the distance is expressed in units of the effective range at resonance $\bar{r}_e$.

parameter, in contrast to what happens for particles undergoing the Efimov effect, for which the three-body parameter is largely set by the finite range of interactions [35,43,44]. A similar situation was numerically observed before for a zero-range model with a momentum cutoff equivalent to a repulsive three-body force [32], while a boundary condition equivalent to an attractive three-body force seems to set a non-zero three-body parameter [33]. We therefore surmise that an attractive three-body force is necessary for the three-body parameter defined by Kartavtsev and Malykh to be non-zero.

The first correction with respect to the zero-range limit (relevant for increasing values of $1/a$) is given by considering the next order in the small-momentum expansion of $|\chi|^2/f_0$. This leads to the STM equation in the effective-range approximation,

$$\left(-\frac{1}{a} + \kappa_0 - \frac{\bar{r}_e}{2}\kappa_0^2\right)s(k) + \int_0^\infty \frac{dk'}{\pi} \frac{yk'}{xk}s(k')\left[2 - \frac{y(k^2+k'^2)+q^2}{xkk'}\right.$$
$$\left.\times \log\left(\frac{y(k^2+k'^2)+q^2+xkk'}{y(k^2+k'^2)+q^2-xkk'}\right)\right] = 0. \quad (30)$$

One can treat the range correction $\frac{\bar{r}_e}{2}\kappa_0^2$ in this last equation as a perturbation with respect to the STM equation in the zero-range limit (see Appendix 4). This yields the corrected energies for the fermion-particle s-wave dimer and the trimer at the first order in the small parameter $\frac{\bar{r}_e}{a}$:

$$E_2 = E_2^{(0)}\left(1 + \frac{\bar{r}_e}{a}\right), \quad (31)$$

$$E_3 = E_3^{(0)}\left(1 + \frac{\bar{r}_e}{a}\langle(\kappa_0 a)^2\rangle\right), \quad (32)$$

where $E_2^{(0)} = y\hbar^2/Ma^2$ and $E_3^{(0)} \propto y\hbar^2/Ma^2$ are the energies obtained in the zero-range limit, and $\langle\cdots\rangle$ denotes the average in the state $|S\rangle$, where $\langle\boldsymbol{k}|S\rangle = S(\boldsymbol{k}) = s(k)\cos(\theta)$ corresponds to the eigenvector $s(k)$ of the uncorrected STM equation, i.e. Eq. (30) in the zero-range limit $\bar{r}_e \to 0$.

Figure 2 shows the spectrum obtained for the various model potentials of Table 1, in the case of a mass ratio $x = 9$, for which only one shallow s-wave induced trimer state exists for

positive scattering length $a$ and vanishes in the three-body threshold at $1/a = 0$. In the range of scattering lengths of the plots, the fermion-particle s-wave dimer binding wave number $q_2$ is nearly identical for all models and is given from Eq. (31) by $q_2 = \sqrt{-2\mu_{23}E_2}/\hbar$. The dashed curve in the left panel represents $-q_2/\sqrt{y}$, which corresponds to the threshold of the dimer+fermion scattering continuum (shaded area).

One can distinguish essentially three regions in these plots. The first region where $\bar{r}_e/a \ll 0.01$ corresponds to the vicinity of the unitary limit where the zero-range approach used by Kartavtsev and Malykh applies. In this region, the energy ratio $E_3/E_2$ between the trimer and dimer is the same for all models and is given within a few tenths of percent by the zero-range limit $1.0457...$ for the considered mass ratio $x = 9$. We note that this universal region is very narrow and would require a fine tuning of the scattering length to be observable in ultracold-atom experiments. As expected, the obtained energy ratio is consistent with the one predicted by the Kartavtsev-Malykh theory with a three-body parameter equal to zero.

The second region, where $\bar{r}_e/a \lesssim 0.01$ is universally described by the scattering length $a$ and the effective range at resonance $\bar{r}_e$ in agreement with Eqs. (31, 32). For the considered mass ratio, we find $\langle(\kappa_0 a)^2\rangle = 2.26\ldots$. This perturbative result, shown as a dashed line in the right panel of Fig. (2), agrees with all models in this region within 0.6%.

In the third region, corresponding to larger values of $\bar{r}_e/a$, the trimer energy becomes non-universal. However, we note that for $\bar{r}_e/a \lesssim 0.1$ it remains nearly the same for all models except the cut-off model. This may be understood by the fact that all these models suppress the wave function within the range $\bar{r}_e$ while hardly affecting it beyond $\bar{r}_e$. This can be seen in Fig. 3. The zero-energy two-body radial wave function $\psi(r)$ for these models roughly approaches what we call the sharp-wave-function limit (dashed curve in Fig. 3) given by:

$$\psi(r) = \begin{cases} 0 & \text{(fully suppressed)} & r \leq r_c, \\ 1 - \frac{r}{a} & \text{(free)} & r > r_c. \end{cases} \tag{33}$$

This wave function has an effective range that reaches the Wigner bound [40, 45, 46], $2r_c\left(1 - \frac{r_c}{a} + \frac{r_c^2}{3a^2}\right)$. It is generically approached by the zero-energy wave function of deep short-range potentials supporting many bound states [35]. It can be exactly obtained from an infinitely deep and narrow potential well located at some distance $r_c$, or from a separable potential with the following form factor:

$$\chi(k) = \left(1 - \frac{r_c}{a}\right)\cos(kr_c) + \frac{\sin(kr_c)}{ka}. \tag{34}$$

The results from this separable potential are shown by the solid black curves in Fig. 2 and are in fair agreement with other models. The cut-off model stands out as an exception, and this can be understood from the fact that its radial wave function is markedly different from the sharp-wave-function limit (33): it features oscillations at large distances (see Fig. 3), which makes its effective range small compared to its true range $1/\Lambda_0$. Interactions with negative effective range, not considered in this study, would even more markedly differ from the sharp-wave-function picture.

## 4 Trimers induced by the p-wave interaction between the fermions

In the following part, we consider the model case where the p-wave interaction is isotropic. We can thus use the symmetry considerations of section 2.2. Since most of the relevant s-wave models are equivalent for $\bar{r}_e/a \lesssim 0.1$, in this section, the s-wave interaction between the fermions and the third particle is described by the Gaussian model.

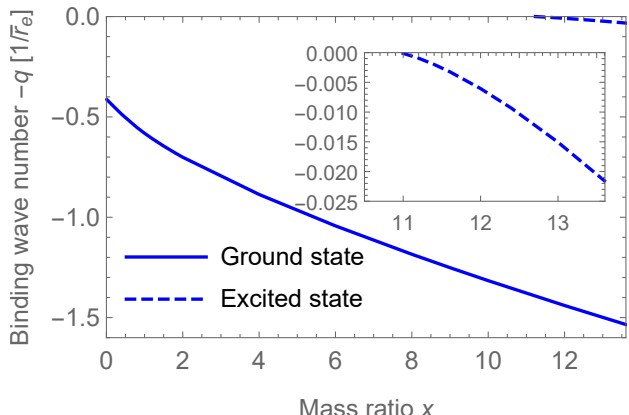

Figure 4: Trimer spectrum in the presence of a p-wave interaction between the fermions, at the doubly resonant limit $1/a \to 0$ and $1/v \to 0$, and as a function of the mass ratio $x$ between the fermions and the third particle. Here, the p-wave interaction is set to the Gaussian model of Table 2 of Appendix 1, with $1/\bar{\alpha} = \bar{r}_e$.

## 4.1 Doubly resonant limit

We first consider simultaneously the s- and p-wave resonant regimes ($1/a = 0$, $1/v = 0$) and in order to avoid the Efimov effect, the spectrum is computed for mass ratios below the critical value $x < x_c$. The limit $1/a = 0$ ensures that there is no shallow fermion-particle s-wave dimer nor s-wave induced trimer, and the limit $1/v = 0$ ensures that there is no shallow fermion-fermion p-wave dimer. One can expect that this regime corresponding to s-wave resonant scattering and in the vicinity of the p-wave resonance is favourable for the occurence of shallow trimers of another type than the KM or Efimov states.

The resulting trimer spectrum is shown in Fig. 4 for the p-wave Gaussian model of Table 2 of Appendix 1. Here, the value of the p-wave interaction range $1/\Lambda_1$ has been set such that $1/\bar{\alpha} = \bar{r}_e$. This arbitrary choice is reasonable for a qualitative understanding of the spectrum, since both lengths are of the order of the range of atomic interactions. We find one shallow trimer for all values of the mass ratio and another one for a mass ratio larger than $x'_c = 10.96\ldots$. Clearly, these trimer states are the consequence of the resonant p-wave interaction that adds to the already present effective attraction between the two fermions due to the fermion-particle s-wave interaction. This is why we adopt the denomination *p-wave induced trimers* for these states. Although the doubly resonant regime is very specific and not directly relevant to experimental systems, it demonstrates the existence of shallow trimers that differ from KM and Efimov states. In what follows, we will change the parameters of the interactions to consider more relevant regimes in view of possible experimental studies of the trimer spectrum.

## 4.2 Weak interaction between the fermions

In this section, we aim at understanding how the p-wave induced trimers emerge from the s-wave induced trimer spectrum of section 3. We thus consider a gradual increase of the p-wave interaction starting from the situation where only the fermion-particle s-wave interaction is present. The mass ratio is fixed at $x = 9$ so that without p-wave interaction there is only one s-wave induced trimer for positive values of the scattering length.

The p-wave attraction is described by the Gaussian, Yamaguchi, and Yamaguchi-squared models detailed in Table (2). Similarly to the study of section 3, we have adjusted the range parameter $\Lambda_1$ to reproduce in all models the same width parameter $\bar{\alpha} = 1/\bar{r}_e$. With this choice

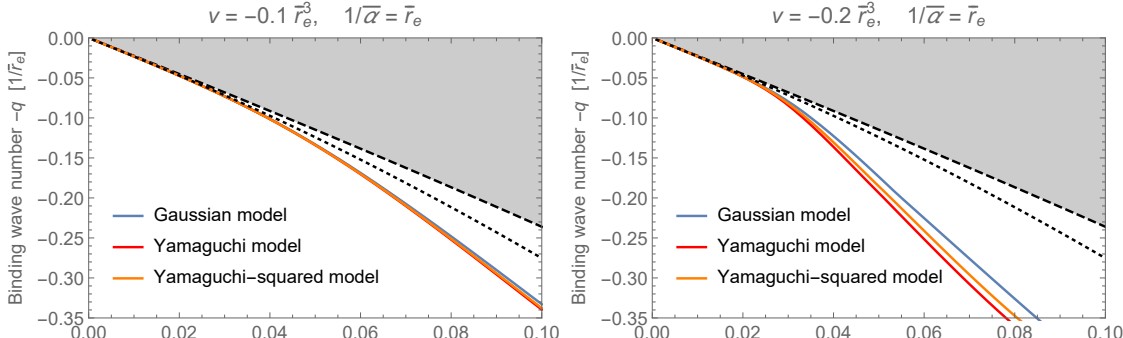

Figure 5: Similar plots to the left panel of Fig. 2 in presence of a weak attraction between the fermions using the models of Table 2 of Appendix 1. Left: the scattering volume is set to $v = -0.1\bar{r}_e^3$; Right: plot for $v = -0.2\bar{r}_e^3$. Dashed curve: limit of the fermion-particle dimer continuum; Dotted curve: s-wave induced trimer without fermion-fermion interaction ($v = 0$) for the Gaussian model, corresponding to the blue curve of Fig. 2.

the diagonal terms of the STM equation are equal for all models in the small momentum limit. The spectrum is plotted in Fig. 5 as a function of $1/a$, for the scattering volumes $v = -0.1\bar{r}_e^3$, and $-0.2\bar{r}_e^3$. For such small scattering volumes, there is no two-body bound state between the two fermions.

As it can be seen, the trimer's binding is significantly strengthened even for these weak fermion-fermion attractions. However, as it could be expected, the KM trimer is not affected by the fermion-fermion attraction in a region of sufficiently large scattering length $a$. This region where the universal theory of Kartavtsev and Malykh applies shrinks with an increasing magnitude of the scattering volume $v$ (compare the left and right panels of Fig. 5). One could understand this situation as follows: for a large scattering length $a$, the KM trimer has a large size (of the order of $a$) and is mostly not affected by the fermion-fermion interaction, since it affects the wave function only within a finite range, much smaller than $a$.

### 4.3 Critical scattering volume at s-wave resonance

For increasing strengths of the fermion-fermion interaction, a radical change occurs in the shallow trimer spectrum as can be seen in Fig. 6. First, as the s-wave induced trimer gets more bound and deviates increasingly from the universal KM state, a new trimer branch appears from the s-wave dimer threshold at finite values of the scattering length. Then, at some critical scattering volume $v_c$, the largest scattering length at which this excited trimer state appears becomes infinite. Concurrently, the ground trimer energy is pushed down so much that it conforms to the KM state limit only at $1/a = 0$. Past this critical strength, the ground trimer becomes borromean - it appears from a negative scattering length - while the excited trimer's threshold remains fixed at $1/a = 0$. As the fermion-fermion attraction is further increased, it is now the excited trimer which follows the KM state limit, in an increasingly wide range of scattering lengths, until this range shrinks again to zero as the scattering volume approaches infinity. For stronger attraction between the fermions, the scattering volume becomes positive and there exists a p-wave two-body bound state, which sets a negative-energy threshold for the occurence of the three-body bound states. Thus, the KM limit does not exist any more in this regime.

We interpret the interplay between the ground and excited trimers as a level repulsion and avoided crossing between the s-wave induced trimer and the p-wave induced trimer states. The analysis of the wave function at $\bar{r}_e^3/v = -3$ shown in Fig. 7 confirms that the s-wave induced

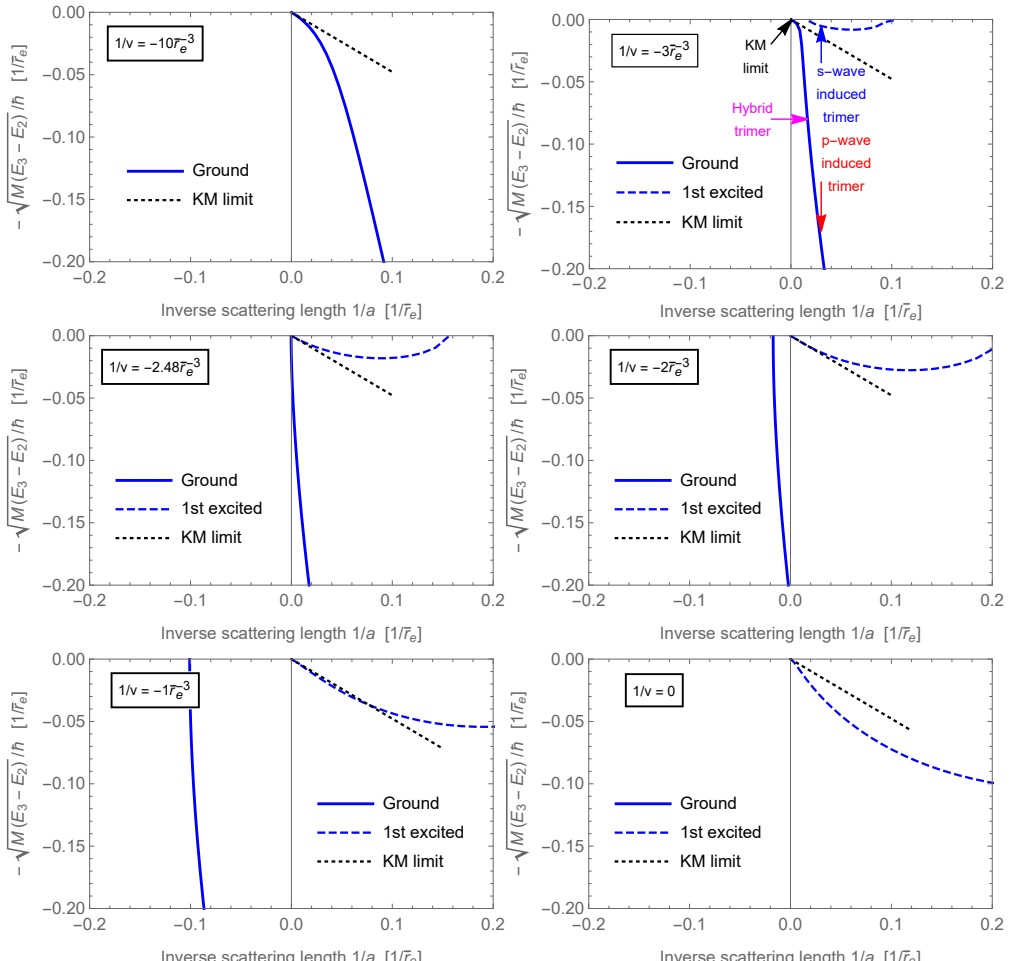

Figure 6: Dependence of the spectrum on the scattering volume $v$ between fermions, for increasing values (in magnitude) of $v$, until a p-wave resonance is reached ($v = -\infty$). The results are obtained with the p-wave Gaussian model, and both the ground (solid curve) and excited (dashed curve) trimer states are shown. In order to distinguish the excited state, the three-body energy is measured from the two-body energy $E_2$ (which is either zero for negative scattering lengths, or the dimer energy for positive scattering lengths) and then converted to a wave number. In this fashion, the Kartavtsev–Malykh zero-range limit corresponds to a straight line shown in black dots. The arrows indicate the states for which the wave functions of Fig. 7 have been computed. The ground state is shown for a wider range of scattering lengths in Fig. 8.

trimer hybridises with the p-wave induced trimer. Before the crossing ($\bar{r}_e/a = 10^{-2}$), $p_0$ is negligible and the wave function $s(k)$ coincides with the solution without p-wave interaction (cross symbols). After the crossing ($\bar{r}_e/a = 3 \times 10^{-2}$), the ground state has a large component $p_0(k)$ and can be considered as the p-wave induced trimer, whereas the excited state has a negligible component $p_0(k)$ and can thus be considered as the s-wave induced trimer. At the crossing ($\bar{r}_e/a = 1.7 \times 10^{-2}$) where there is still one trimer state, the component $p_0(k)$ is not negligible as a consequence of the hybridisation. Thus for increasing values of $1/a$ the ground branch of the s-wave induced trimer continuously transforms into the p-wave induced trimer.

As can be seen in Fig. 8, the regime where the ground-state trimer is borromean is not universal: different models with the same parameters $v$ and $\bar{\alpha}$ give different results. Nevertheless, these results remain qualitatively consistent. Figure 8 shows the dependence on the

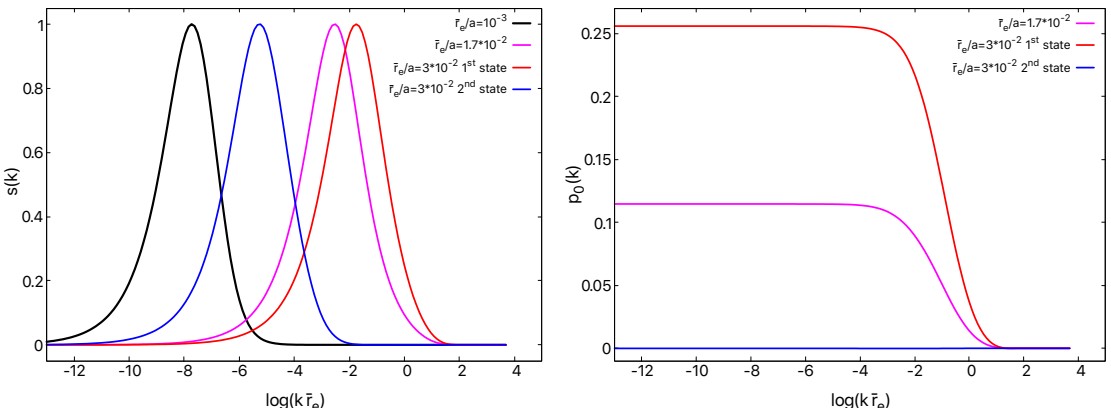

Figure 7: Components $s$ (left panel) and $p_0$ (right panel) of the wave function in Eq. (23) in vicinity of the crossing between the s-wave and the p-wave induced trimers for a mass ratio $x = 9$ at $\bar{r}_e^3 / v = -3$ (see the top right panel of Fig. 6). The component $p_2$ is negligible and not plotted (see section 5.1). For $\bar{r}_e/a = 10^{-3}$ (well before the crossing), the s-wave induced trimer (black) is indistinguishable from the KM state. For $\bar{r}_e/a = 1.7 \times 10^{-2}$, the s-wave induced trimer (magenta) hybridises with the (virtual) p-wave induced trimer. At $\bar{r}_e/a = 3 \times 10^{-2}$ (after the crossing) there are two trimers: the first state (ground state) shown in red and the second state (excited state) shown in blue.

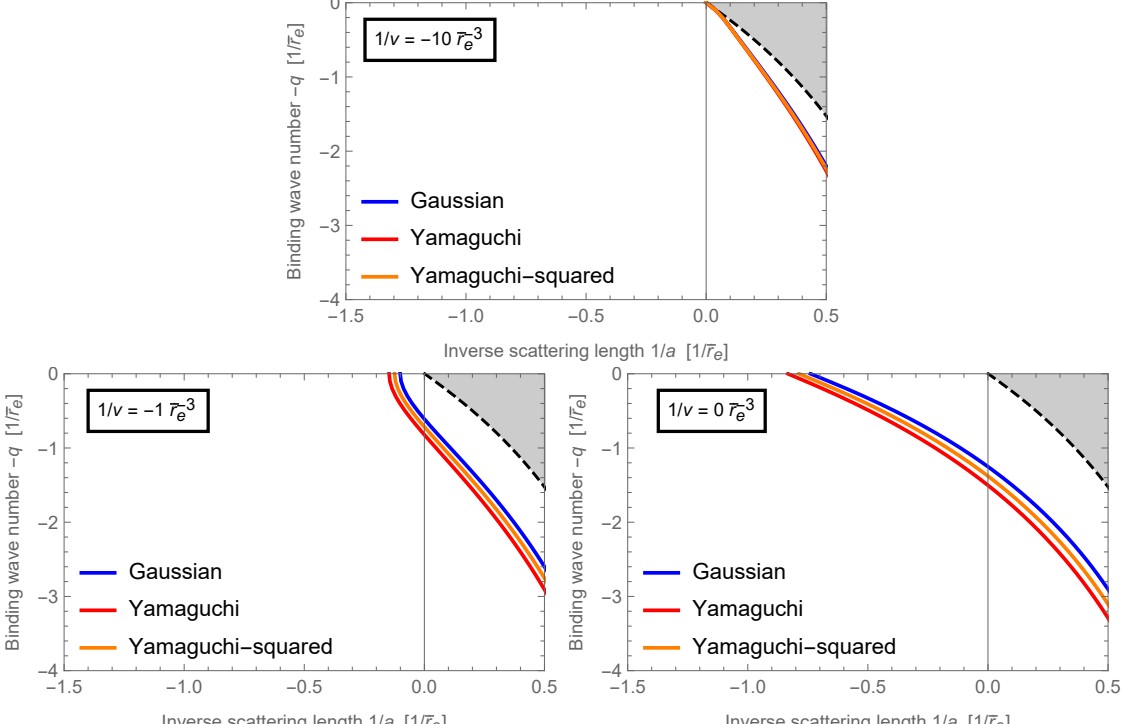

Figure 8: Dependence of the spectrum on the scattering volume $v$ between fermions, for increasing values (in magnitude) of $v$, until a p-wave resonance is reached ($v = -\infty$). The plots are similar to the left panel of Fig. 2, although the different curves now correspond to different models of the p-wave interaction between the fermions. The value of the width parameter $\bar{\alpha}$ is fixed to $1/\bar{r}_e$ in all models. On the scale of these plots, only the ground trimer is visible, the excited trimer being indistinguishable from the dimer (dashed curve).

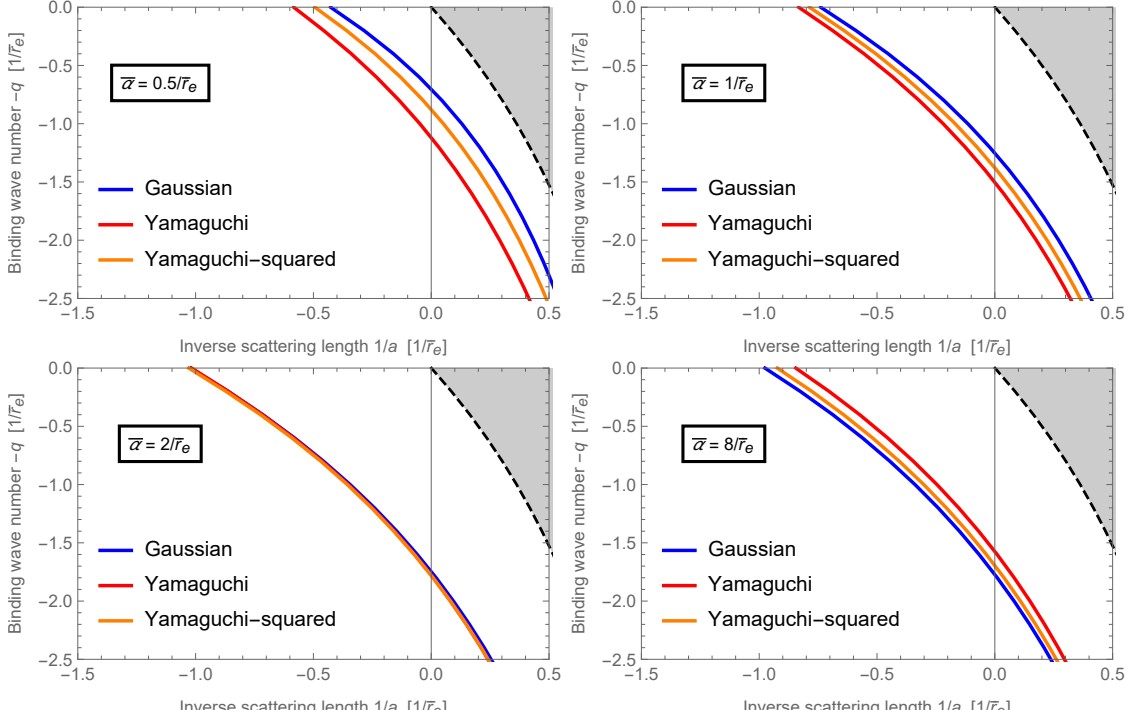

Figure 9: At the p-wave resonance $v = \infty$. Plots similar to Fig. 8 where, instead of the scattering volume $v$, the width parameter $\bar{\alpha}$ is varied as indicated in the box of each panel. Note that the second panel is the same as the last panel of Fig. 8.

scattering volume for a fixed effective range. There is also a dependence on the effective range. This is shown in Fig. 9, at the fixed scattering volume $v = \infty$ corresponding to the p-wave resonance, i.e. the occurence of a p-wave two-body bound state between the two fermions. One can see more marked differences between the models for a p-wave width parameter $\bar{\alpha}$ that is small compared to $1/\bar{r}_e$, meaning that we consider a p-wave potential of large range with respect to the s-wave potential ($\Lambda_1 \ll \Lambda_0$). At the particular value $1/\bar{\alpha} = \frac{1}{2}\bar{r}_e$, all models appear to coincide, although this is specific to this value and we have no particular explanation for this coincidence.

Let us now consider the critical scattering volume at which the fermion-fermion interaction is sufficiently attractive for supporting a borromean p-wave induced trimer. This occurs at the s-wave resonance ($1/a = 0$) of the fermion-particle interaction, where the s-wave dimer appears. As shown in Fig. (6), for the mass ratio $x = 9$, using the p-wave Gaussian model, we find $1/v = -2.48/\bar{r}_e^3$ with the choice $\bar{\alpha} = 1/\bar{r}_e$.

Figure 10 shows the ground-state trimer energy at the s-wave unitarity as a function of the inverse scattering volume for three different models satisfying $\bar{\alpha} = 1/\bar{r}_e$. The critical scattering volume $v_c$ is indicated by an arrow for each model. As it can be seen, the critical scattering volume is not universal, although it is close to $1/v_c \approx -2.5/\bar{r}_e^3$ for the three models.

The critical scattering volume itself depends on the interaction range between the fermions, which is of the order of $1/\bar{\alpha}$. This dependence is shown in the left panel of Fig. 11 for the mass ratio $x = 9$. Again, the results are not model-independent but they are sufficiently close to draw general conclusions. One can see that the critical inverse scattering volume vanishes for $1/\bar{\alpha} \approx \bar{r}_e/34$ and has a maximum magnitude around $1/\bar{\alpha} \approx \bar{r}_e/10$. This means that the p-wave induced trimer usually exists, unless the range $1/\bar{\alpha}$ of the fermion-fermion interaction happens to be more than 30 times smaller than the range of the particle-fermion interaction. For a fermion-fermion interaction range $1/\bar{\alpha}$ of the same order as the fermion-particle interaction

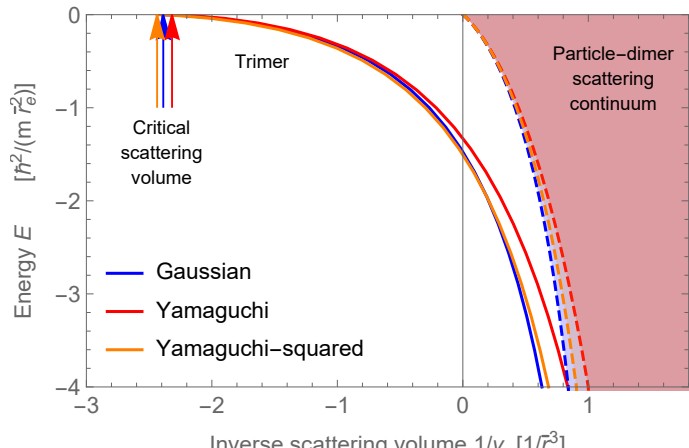

Figure 10: Three-body spectrum as a function of the inverse scattering volume $1/v$ between the two fermions, for a fermion-particle interaction at the s-wave unitarity ($a = \infty$) and mass ratio $x = 9$. The solid curves indicate the trimer energy for different models and the dashed curves indicate the p-wave fermion-fermion dimer energy. The arrows indicate the critical scattering volume $v_c$ at which the trimer appears from the three-body scattering threshold. This critical scattering volume corresponds to the value beyond which the trimer is borromean with respect to the fermion-particle dimer, as seen in Fig. 8.

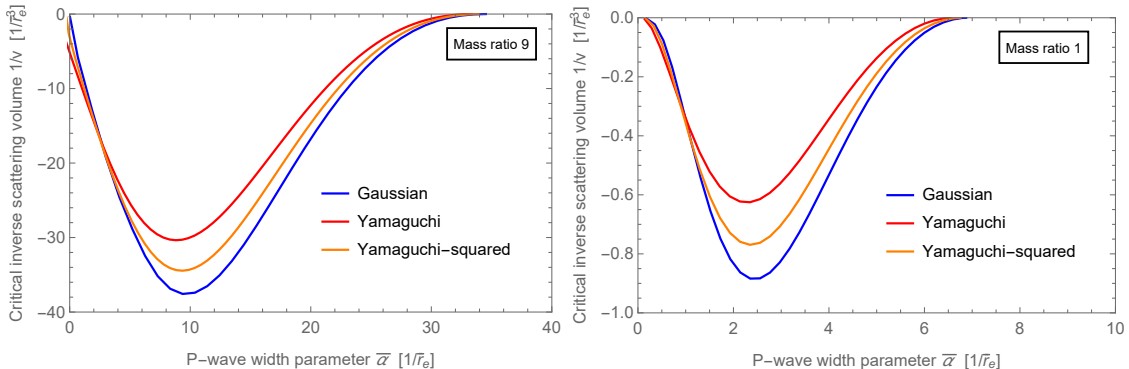

Figure 11: Critical inverse scattering volume $1/v_c$, as a function of the width parameter $\bar{\alpha}$. Left panel: for a mass ratio 9. Right panel: for a mass ratio 1.

range $\bar{r}_e$, the critical scattering volume required to observe the p-wave induced trimer is about $-\bar{r}_e^3/10$, which means that the fermion-fermion interaction remains modest and far from the p-wave resonance $v\bar{\alpha}^{-3} \gg 1$.

So far, we have looked at the specific mass ratio $x = 9$, since in the absence of fermion-fermion interaction, KM trimers only exist for $M/m > x_1 = 8.17260\ldots$. However, as we pointed out in Section 4.1, in the presence of fermion-fermion interaction, the p-wave induced trimers may exist for any mass ratio. The right panel of Fig. 11 shows the critical scattering volume for a mass ratio 1 (all three particles having the same mass, such as identical atoms in different spin states). The figure shows that the p-wave induced trimer still exists, however the required critical scattering volume is larger than for a mass ratio of 9. For a width parameter $\bar{\alpha}$ of the order of $1/\bar{r}_e$, the scattering volume must be larger (in absolute value) than $3.5\bar{r}_e^3$, which is closer to the p-wave resonance.

Finally, we should mention that for small positive scattering lengths or volumes, the models used in this study lead to the presence of a series of deep trimer states that emerge successively

from the dimer threshold. These deep trimers are due to the usual fact that strong attractive interactions bind more strongly three particles than two particles. Unlike the shallow trimers discussed above, these trimers are very model-dependent. Furthermore, for such deep states the models do not correpond to any accurate description of a real system. For these reasons, we do not consider these deep trimers. It is nonetheless interesting to note that as the system approaches the p-wave resonance, the excited state shown in Fig. (6) at some point merges with the lowest of these deep states, in a region of small positive scattering length outside the range of Fig. (6) . Namely, the scattering length at which the excited state disappears in the dimer threshold meets the scattering length at which the deep state appears from the threshold, and the two states become a single state lying below the dimer threshold.

# 5 Non isotropic p-wave interaction

In the presence of a magnetic field along the $z$ direction, the p-wave interaction is not isotropic: a phenomenon observed in experiments [47, 48]. This results in a difference between the scattering amplitude $f_{1,0}$ and the scattering amplitudes $f_{1,1} = f_{1,-1}$ thus leading to a splitting of the isotropic trimer branches (degeneracy of order three) into two branches. In principle, this makes the formalism more involved as the total orbital momentum of the three particles is no more a good quantum number. However the magnetic quantum number of the total orbital momentum and the parity are still good quantum numbers of the system. The S function is then the superposition of odd momentum states with the desired projection on the $z$-axis ($M = 0, \pm 1$). The direct diagonalisation of the STM equation is then a bit tedious.

As we show in what follows, we will use an accurate approximation of the wave function that permits us to greatly simplify the problem in a perturbative approach.

## 5.1 S-wave coupling approximation

For an isotropic p-wave interaction, we have observed that the wave functions $\mathbf{P}(\mathbf{k})$ obtained numerically predominantly consist of the s-wave state for the relative motion between the impurity and the pair of identical fermions in a p-wave state, meaning that $|p_2(k)| \ll |p_0(k)|$ and thus

$$\mathbf{P}(\mathbf{k}) \sim p_0(k)\hat{\mathbf{e}}_z \,. \tag{35}$$

Physically, this can be understood by the fact that the coupling with the d-wave state is suppressed due to the d-wave centrifugal barrier. In what follows, Eq. (35) will be called the *s-wave coupling approximation*.

Inspection of the STM equations (24-26) shows that this corresponds to neglecting the terms involving the kernel $L_2$. One can verify indeed that in the region $k\bar{r}_e < 1$,

$$\left| \frac{\int_0^\infty dk' k'^2 L_2(k,k')s(k')}{\int_0^\infty dk' k'^2 L_0(k,k')s(k')} \right| \ll 1 \,. \tag{36}$$

This last property, which depends in principle on the particular form of the function $s(k')$, is very well satisfied for the solutions of the STM equations, which decrease quickly for $k\bar{r}_e \gtrsim 1$. This can be explained qualitatively by the fact that in the domain $k\bar{r}_e, k'\bar{r}_e \ll 1$, the kernel $L_2$ may be approximated by its zero-range limit where the functions $\chi$ and $\phi$ are replaced by the unit number:

$$L_2(k,k') \sim 3 \int_{-1}^1 du \frac{k\left(3u^2 - 1\right) + k'u}{k^2 + yk'^2 + kk'u + q^2} \,, \tag{37}$$

which is exactly zero in the limit $k^2 + y k'^2 + q^2 \gg k k'$. This limit is indeed always verified in the domain $k \ll q$, where the function $p_0(k)$ is almost constant. In particular, it is satisfied for sufficiently deep states ($q \bar{r}_e$ of the order of the unity), which is typically the case for the p-wave induced trimers. For s-wave induced trimers, after a plateau the function $p_0$ decreases quickly for $k > q$ (see for example typical shapes of this function in Fig. 7), and the s-wave coupling approximation is thus relevant to compute these states. For p-wave induced trimers, we observe that $p_0(k)$ is almost constant for $k \lesssim \Lambda_0 \sim \Lambda_1$, even near the threshold. Outside this plateau, the s-wave coupling approximation is not satisfied in general but $p_0(k)$ becomes negligible. Hence, we conclude that the s-wave coupling approximation has a very wide regime of validity.

In the s-wave coupling approximation, the matrix STM equations can be written as

$$\frac{|\chi(i\kappa_0)|^2}{f_0(i\kappa_0)} s(k) + \int_0^\infty \frac{dk'}{\pi} k'^2 \Big[ L(k, k') s(k') + 3 L_0(k, k') p_0(k') \Big] = 0 \,, \tag{38}$$

$$3 \frac{|\kappa_1 \phi(i\kappa_1)|^2}{2 y f_1(\kappa_1)} p_0(k) - \int_0^\infty \frac{dk'}{\pi} k'^2 L_0(k', k)^* s(k') = 0 \,. \tag{39}$$

In the next section, we will use the s-wave coupling approximation to treat the anisotropic p-wave interaction perturbatively.

## 5.2 Perturbative approach

For a given value of the quantum number $M$, we rewrite Eq. (7) in the form

$$\frac{[\kappa_1 \phi(i\kappa_1)]^2 P(\boldsymbol{k})}{4\pi f_{1,M}(i\kappa_1)} - 2y \int \frac{d^3 k'}{(2\pi)^3} \left[ \langle \phi | \boldsymbol{k}' + \frac{\boldsymbol{k}}{2} \rangle \frac{(\boldsymbol{k}' + \frac{\boldsymbol{k}}{2}) S(\boldsymbol{k}') \langle \frac{\boldsymbol{k}'}{2y} + \boldsymbol{k} | \chi \rangle}{k'^2 + y k^2 + q^2 + \boldsymbol{k} \cdot \boldsymbol{k}'} \right] + \delta \boldsymbol{X}_M = 0 \,. \tag{40}$$

In Eq. (40), we have introduced the perturbation $\delta \boldsymbol{X}_M$ with respect to the formally isotropic STM equations where we have set $g \equiv g_M$, $\phi \equiv \phi_M$ and $f_1(k) \equiv f_{1,M}(k)$:

$$\delta \boldsymbol{X}_M = \sum_{m=-1}^{1} \left\{ \frac{[\kappa_1 \phi_m(i\kappa_1)]^2}{f_{1,m}(i\kappa_1)} - \frac{[\kappa_1 \phi_M(i\kappa_1)]^2}{f_{1,M}(i\kappa_1)} \right\} P_m \hat{\boldsymbol{e}}_m \,. \tag{41}$$

At the lowest order, i.e. if one neglects the perturbation, the s-wave coupling approximation gives the form of the $S$ and $\boldsymbol{P}$ functions:

$$S(\boldsymbol{k}) = \hat{e}_{\boldsymbol{k}} \cdot \hat{e}_M s(k) \,; \quad P_m(\boldsymbol{k}) = p_0(k) \delta_{M,m} \,. \tag{42}$$

We can then use the perturbation formalism of Appendix 4. The unperturbed eigenvector is

$$\langle \boldsymbol{k} | \lambda_0 \rangle = [s(k)(\hat{e}_{\boldsymbol{k}} \cdot \hat{e}_M), p_0(k) \delta_{M,m}] \tag{43}$$

and the shift in energy at the first order of the perturbation is obtained from the matrix element

$$\langle \lambda_0 | \mathcal{M}^{(1)} | \lambda_0 \rangle = \sum_{m=-1}^{1} \int \frac{d^3 k}{(2\pi)^3} (\hat{e}_m^* \cdot \delta \boldsymbol{X}_M) P_m(\boldsymbol{k}) \,. \tag{44}$$

Using Eq. (42), one finds that this term is exactly zero and thus that at the first order of perturbation, the formally isotropic model gives the branches of the spectrum corresponding to the magnetic number $m$.

To conclude this section, using the s-wave coupling approximation, the spectrum indexed by the quantum number $M$ is obtained by considering for each value of $M$ the system of equations (38,39) where $f_1(i\kappa_1)$ is replaced by $f_{1,M}(i\kappa_1)$. The results derived in the isotropic case can thus be directly used to determine the spectrum for an anisotropic p-wave interaction.

## 6 Experimental observation

The three-body spectrum presented in this study can be probed experimentally in ultra-cold experiments with standard techniques used for the study of Efimov trimers. The traditional and most straightforward technique consists in preparing an ultra-cold mixture of atoms in a certain state of interaction, and measure the three-body losses by imaging the cloud of atoms [6, 7, 9–11]. The variation of the loss rate as a function of a parameter controlling the interactions should reveal features related to the spectrum. For instance, near the appearance threshold of the Borromean trimer presented above, a peak in the losses by three-body recombination is expected. The detailed shape of this peak has not been addressed here.

Alternatively, depending on the considered atomic species, it is possible to use the radio-frequency spectroscopy technique, which consists in preparing the atoms in different spin states than those for which the interactions support the trimers of interest described in this study, and induce a spin transition which target those trimers, and thus measure their energy [15, 20, 21]. Such measurements could directly test our theoretical predictions.

However, the first challenge in such experiments is to find and prepare two atomic species that have a suitable mass ratio and whose two-body s-wave and p-wave interactions happen to be, or can be controlled to be, in a region of scattering length and scattering volume where interesting trimers states are present. This work shows that the original constraint on the mass ratio for the KM states, which limits the choice mostly to mixtures of chromium and lithium atoms, can be relaxed provided there is a sufficiently strong p-wave interaction between the two fermionic atoms. Finding the most promising candidates requires a precise knowledge of the interspecies s-wave resonances of mass imbalanced mixtures, as well as their intraspecies p-wave resonances. One possible candidate is the potassium-40 two-species mixture at a magnetic field near 200G, which features both an s-wave and a p-wave resonance [50].

## 7 Conclusion

In this work, we have clarified how the universal trimer bound states of two identical fermions and one particle discovered by Kartavtsev and Malykh occur in a realistic setting where finite-range interactions are present between all particles. We found that in the absence of a fermion-fermion dimer, the Kartavtsev–Malykh universal trimers are almost always present for a sufficiently large scattering length between the fermions and the particle. However, this range of scattering length is very narrow, making the experimental observation of these universal states challenging. The trimers extend away from this universal region to smaller scattering lengths, where they become significantly more bound. Moreover, we found that the spectrum is enriched by up to two additional trimers for sufficiently strong attraction between the fermions. These trimers are borromean, i.e. they can exist even though the interactions are not sufficiently attractive to bind the two fermions or bind the particle with one of the two fermions. Although this extended spectrum is not strictly universal, it follows a generic scenario as a function of the low-energy parameters of the interactions. This scenario results from an avoided crossing between trimers induced only by the s-wave interaction between the fermions and the particle, and trimers induced by the p-wave interaction between the two fermions. A striking feature of the borromean trimers is that their ground state exists for any mass ratio, unlike the universal trimer states whose existence are limited to mass ratios larger than $x_1 = 8.17260\ldots$. This dramatic enhancement of the range of existence, in terms of both interaction and mass ratio, makes these trimers much more accessible to experimental observation.

It is interesting to mention that a shallow borromean trimer in the same symmetry sector

$J^{\pi} = 1^-$ has also been predicted in the case of three fully polarised fermions experiencing a p-wave resonant pairwise interaction [49]. We should also note that our conclusions are limited to the case of single-channel interactions between the particles. These would describe the so-called open-channel-dominated resonances in real systems. A more general description of resonances would require at least two channels.

## Acknowledgments

C. S. and P. N. acknowledge support from the JSPS Grants-in-Aid for Scientific Research on Innovative Areas (No. JP18H05407).

## A    Separable potential for different partial waves

A separable potential acting on the $\ell$th partial wave has the general form:

$$\hat{V} = \sum_{m=-\ell}^{\ell} \lambda_{\ell,m} |\Phi_{\ell,m}\rangle \langle \Phi_{\ell,m}|, \tag{45}$$

with

$$\langle \boldsymbol{k}|\Phi_{\ell,m}\rangle = \Phi_{\ell,m}(\boldsymbol{k}) = k^{\ell} \phi_{\ell,m}(k)\sqrt{4\pi} Y_{\ell,m}(\hat{\boldsymbol{k}}), \tag{46}$$

where $Y_{\ell,m}$ are the spherical harmonics and $\hat{\boldsymbol{k}}$ is the unit vector $\boldsymbol{k}/k$.

The action of this separable potential on a wave function $\Psi$ is given in momentum representation by $\langle \boldsymbol{k}|\hat{V}|\Psi\rangle$, i.e.

$$\sum_{m=-\ell}^{\ell} \lambda_{\ell m} \int d^3 k' \Phi_{\ell,m}^*(\boldsymbol{k}') \Phi_{\ell,m}(\boldsymbol{k}) \langle \boldsymbol{k}'|\Psi\rangle$$

$$= \sum_{m=-\ell}^{\ell} 4\pi \lambda_{\ell,m} k^{\ell} \phi_{\ell,m}(k) Y_{\ell,m}(\hat{\boldsymbol{k}}) \int_0^{\infty} dk' k'^{\ell} \phi_{\ell,m}^*(k') \underbrace{\int d^2 \hat{k}' Y_{\ell,m}^*(\hat{\boldsymbol{k}}') \langle \boldsymbol{k}'|\Psi\rangle}_{\Psi_{\ell,m}(k')}, \tag{47}$$

where $\Psi_{\ell,m}$ is the partial wave $(\ell, m)$ of the wave function $\Psi$. Thus we see that the potential only acts on that partial wave.

For the $s$ wave ($\ell = 0$) we have $Y_{0,0}(\hat{\boldsymbol{k}}) = 1/\sqrt{4\pi}$, and thus,

$$\hat{V} = \lambda_{0,0} |\phi_{0,0}\rangle \langle \phi_{0,0}|, \tag{48}$$

which is the form of Eq. (2) with $\lambda_{0,0} \equiv \frac{2\pi\hbar^2}{\mu_{23}}\xi$ and $\phi_{0,0} \equiv \chi$.

For the $p$ wave ($\ell = 1$) we have $Y_{1,m}(\hat{\boldsymbol{k}}) = \sqrt{\frac{3}{4\pi}}\hat{\boldsymbol{k}} \cdot \hat{\boldsymbol{e}}_m$, and thus,

$$\hat{V} = \sum_{m=-1}^{1} \lambda_{1,m} |\Phi_{1,m}\rangle \langle \Phi_{1,m}|, \tag{49}$$

with

$$\Phi_{1,m}(\boldsymbol{k}) = k\phi_{1,m}(k)\sqrt{3}\hat{\boldsymbol{k}} \cdot \hat{\boldsymbol{e}}_m, \tag{50}$$

which is the form of Eq. (3), with $\lambda_{1,m} \equiv \frac{2\pi\hbar^2}{\mu_{12}}g_m$ and $\Phi_{1,m}(\boldsymbol{k}) \equiv \Phi_m(\boldsymbol{k})$, i.e. $\phi_{1,m}(k) \equiv \phi_m(k)/\sqrt{3}$.

The Tables (1) and (2) give the different parameters of the potentials used in this paper.

Table 1: Separable potential models used for the s-wave interaction $V$ between a fermion and the particle. For each model, the table provides the explicit expression of the form factor $\chi$, the s-wave scattering amplitude $f_0$, and the parameter $\bar{r}_e$ corresponding to the s-wave effective range at resonance.

| s-wave model | $\chi(k)$ | $\lvert\chi(i\kappa)\rvert^2/f_0(i\kappa)$ | $\bar{r}_e$ |
|---|---|---|---|
| Gaussian | $\exp(-k^2/\Lambda_0^2)$ | $\frac{1}{a} - \kappa e^{\frac{2\kappa^2}{\Lambda_0^2}}\operatorname{erfc}\left(\frac{\sqrt{2}\kappa}{\Lambda_0}\right)$ | $\frac{4\sqrt{\frac{2}{\pi}}}{\Lambda_0}$ |
| Yamaguchi | $\frac{1}{1+k^2/\Lambda_0^2}$ | $\frac{1}{a} - \frac{\kappa\Lambda_0(\kappa+2\Lambda_0)}{2(\kappa+\Lambda_0)^2}$ | $\frac{3}{\Lambda_0}$ |
| Yamaguchi-squared | $\left[\frac{1}{1+k^2/\Lambda_0^2}\right]^2$ | $\frac{1}{a} - \frac{\kappa\Lambda_0\left(5\kappa^3+20\kappa^2\Lambda_0+29\kappa\Lambda_0^2+16\Lambda_0^3\right)}{16(\kappa+\Lambda_0)^4}$ | $\frac{35}{8\Lambda_0}$ |
| Cut-off | $\begin{cases}1 & k\le\Lambda_0 \\ 0 & k>\Lambda_0\end{cases}$ | $\frac{1}{a} - \frac{2\kappa\tan^{-1}\left(\frac{\Lambda_0}{\kappa}\right)}{\pi}$ | $\frac{4}{\pi\Lambda_0}$ |

Table 2: Separable potential models used for the p-wave interaction $U$ between the two fermions. For each model, the table provides the explicit expression of the form factor $\phi$, the p-wave scattering amplitude $f_1$, and the width parameter $\bar{\alpha}$ corresponding to the inverse of the effective range at the p-wave resonance.

| p-wave model | $\phi(k)$ | $\lvert\kappa\phi(i\kappa)\rvert^2/f_1(i\kappa)$ | $\bar{\alpha}$ |
|---|---|---|---|
| Gaussian | $\exp(-k^2/\Lambda_1^2)$ | $\frac{1}{v} - \frac{\kappa^2\Lambda_1}{\sqrt{2\pi}} + \kappa^3 e^{\frac{2\kappa^2}{\Lambda_1^2}}\operatorname{erfc}\left(\frac{\sqrt{2}\kappa}{\Lambda_1}\right)$ | $\frac{\Lambda_1}{\sqrt{2\pi}}$ |
| Yamaguchi | $\frac{1}{1+k^2/\Lambda_1^2}$ | $\frac{1}{v} - \frac{\kappa^2\Lambda_1^3}{2(\kappa+\Lambda_1)^2}$ | $\frac{\Lambda_1}{2}$ |
| Yamaguchi squared | $\left[\frac{1}{1+k^2/\Lambda_1^2}\right]^2$ | $\frac{1}{v} - \frac{\kappa^2\Lambda_1^3\left(\kappa^2+4\kappa\Lambda_1+5\Lambda_1^2\right)}{16(\kappa+\Lambda_1)^4}$ | $\frac{5\Lambda_1}{16}$ |
| Cut-off | $\begin{cases}1 & k\le\Lambda_1 \\ 0 & k>\Lambda_1\end{cases}$ | $\frac{1}{v} - \frac{2}{\pi}\kappa^2\left(\Lambda_1 - \kappa\tan^{-1}\left(\frac{\Lambda_1}{\kappa}\right)\right)$ | $\frac{2}{\pi}\Lambda_1$ |

# B   Derivation of the STM equations

Here we derive the STM equations (6-7).

First, we use the separable forms (2) and (3) of the interactions $\hat{V}$ and $\hat{U}$ in the Schrödinger equation (1). This gives:

$$\left(\frac{1}{2}k_1^2 + \frac{1}{2}k_2^2 + \frac{x}{2}k_3^2 + q^2\right)\langle\{k_i\}|\Psi\rangle - 12\pi\sum_{m=-1}^{1}\langle k_{12}|\Phi_m\rangle P_m(k_3)$$
$$- 4\pi y\langle k_{23}|\chi\rangle S(k_1) + 4\pi y\langle k_{31}|\chi\rangle S'(k_2) = 0, \qquad (51)$$

where $x = M/m$ is the mass ratio, and $y$ is the short-hand notation for $(1+x)/2$. The relative wave functions $S$ and $P_m$ are defined by Eqs. (4-5), and $S'$ is given by:

$$S'(k_2) = -\xi\int\frac{d^3k_{31}}{(2\pi)^3}\langle\chi|k_{31}\rangle\langle\{k_i\}|\Psi\rangle. \qquad (52)$$

From the antisymmetry of the wave function under the exchange of the fermionic particles 1 and 2, i.e. $\langle k_2, k_1, k_3|\Psi\rangle = -\langle k_1, k_2, k_3|\Psi\rangle$, it is easy to check from Eq. (52) that $S'(k) = -S(k)$. The three-body wave function $\langle\{k_i\}|\Psi\rangle/4\pi$ is thus given by:

$$\frac{3\sum_m\langle k_{12}|\Phi_m\rangle P_m(k_3) + y(\langle k_{23}|\chi\rangle S(k_1) - \langle k_{31}|\chi\rangle S(k_2))}{\frac{1}{2}k_1^2 + \frac{1}{2}k_2^2 + \frac{x}{2}k_3^2 + q^2}. \qquad (53)$$

Inserting this expression in the definitions of $S$ and $P_m$, Eqs. (4-5), one obtains:

$$-\frac{S(\boldsymbol{k}_1)}{4\pi\xi} = 3\int \frac{d^3k_{23}}{(2\pi)^3}\langle\chi|\boldsymbol{k}_{23}\rangle \frac{\sum_m\langle\boldsymbol{k}_{12}|\Phi_m\rangle P_m(\boldsymbol{k}_3)}{\frac{1}{2}k_1^2 + \frac{1}{2}k_2^2 + \frac{x}{2}k_3^2 + q^2}$$
$$+ y\int \frac{d^3k_{23}}{(2\pi)^3}\langle\chi|\boldsymbol{k}_{23}\rangle \frac{\langle\boldsymbol{k}_{23}|\chi\rangle S(\boldsymbol{k}_1) - \langle\boldsymbol{k}_{31}|\chi\rangle S(\boldsymbol{k}_2)}{\frac{1}{2}k_1^2 + \frac{1}{2}k_2^2 + \frac{x}{2}k_3^2 + q^2}, \tag{54}$$

$$-\frac{P_m(\boldsymbol{k}_3)}{4\pi g_m} = 3\int \frac{d^3k_{12}}{(2\pi)^3}\langle\Phi_m|\boldsymbol{k}_{12}\rangle \frac{\sum_{m'}\langle\boldsymbol{k}_{12}|\Phi_{m'}\rangle P_{m'}(\boldsymbol{k}_3)}{\frac{1}{2}k_1^2 + \frac{1}{2}k_2^2 + \frac{x}{2}k_3^2 + q^2}$$
$$+ y\int \frac{d^3k_{12}}{(2\pi)^3}\langle\Phi_m|\boldsymbol{k}_{12}\rangle \frac{\langle\boldsymbol{k}_{23}|\chi\rangle S(\boldsymbol{k}_1) - \langle\boldsymbol{k}_{31}|\chi\rangle S(\boldsymbol{k}_2)}{\frac{1}{2}k_1^2 + \frac{1}{2}k_2^2 + \frac{x}{2}k_3^2 + q^2}. \tag{55}$$

From the translational invariance of the system, the total momentum can be set to zero:

$$\boldsymbol{k}_1 + \boldsymbol{k}_2 + \boldsymbol{k}_3 = 0. \tag{56}$$

It follows that the three vectors $\boldsymbol{k}_1$, $\boldsymbol{k}_2$, and $\boldsymbol{k}_3$ can be expressed in terms of two Jacobi vectors $\boldsymbol{k}_{ij}$ and $\boldsymbol{k}_k$:

$$\boldsymbol{k}_1 = \boldsymbol{k}_{31} - \frac{x}{1+x}\boldsymbol{k}_2 = -\boldsymbol{k}_{12} - \frac{1}{2}\boldsymbol{k}_3, \tag{57}$$

$$\boldsymbol{k}_2 = -\boldsymbol{k}_{23} - \frac{x}{1+x}\boldsymbol{k}_1 = \boldsymbol{k}_{12} - \frac{1}{2}\boldsymbol{k}_3, \tag{58}$$

$$\boldsymbol{k}_3 = +\boldsymbol{k}_{23} - \frac{1}{1+x}\boldsymbol{k}_1 = -\boldsymbol{k}_{31} - \frac{1}{1+x}\boldsymbol{k}_2. \tag{59}$$

In particular, the denominator in Eqs. (54-55) admits the following expressions:

$$\frac{1}{2}k_1^2 + \frac{1}{2}k_2^2 + \frac{x}{2}k_3^2 + q^2 = yk_{23}^2 + k_1^2\frac{2x+1}{4y} + q^2 \tag{60}$$

$$= k_{12}^2 + \frac{2x+1}{4}k_3^2 + q^2. \tag{61}$$

Now, one can see that the term proportional to $S(\boldsymbol{k}_1)$ in the right-hand side of Eq. (54) can be factored with the left-hand side. Similarly, the term proportional to $P_{m'}(\boldsymbol{k}_3)$ in the right-hand side of Eq. (55) can be shown to be proportional to $P_m(\boldsymbol{k}_3)$, by performing the integration over the orientation of $\boldsymbol{k}_{12}$ and using the relation $\int d^2\hat{\boldsymbol{k}}_{12}(\hat{\boldsymbol{k}}_{12}\cdot\hat{\boldsymbol{e}}_m)(\hat{\boldsymbol{k}}_{12}\cdot\hat{\boldsymbol{e}}_{m'}) = \frac{4\pi}{3}\delta_{m,m'}$. It can therefore be factored with the left-hand side. Using Eq. (60) in Eq. (54) and Eq. (61) in Eq. (55), one obtains:

$$\frac{S(\boldsymbol{k}_1)}{4\pi\Xi_{0,0}} = -3\int \frac{d^3k_{23}}{(2\pi)^3}\langle\chi|\boldsymbol{k}_{23}\rangle \frac{\sum_m\langle\boldsymbol{k}_{12}|\Phi_m\rangle P_m(\boldsymbol{k}_3)}{\frac{1}{2}k_1^2 + \frac{1}{2}k_2^2 + \frac{x}{2}k_3^2 + q^2}$$
$$+ y\int \frac{d^3k_{23}}{(2\pi)^3}\langle\chi|\boldsymbol{k}_{23}\rangle \frac{\langle\boldsymbol{k}_{31}|\chi\rangle S(\boldsymbol{k}_2)}{\frac{1}{2}k_1^2 + \frac{1}{2}k_2^2 + \frac{x}{2}k_3^2 + q^2}, \tag{62}$$

$$\frac{P_m(\boldsymbol{k}_3)}{4\pi\Xi_{1,m}} = y\int \frac{d^3k_{12}}{(2\pi)^3}\langle\Phi_m|\boldsymbol{k}_{12}\rangle \frac{\langle\boldsymbol{k}_{31}|\chi\rangle S(\boldsymbol{k}_2) - \langle\boldsymbol{k}_{23}|\chi\rangle S(\boldsymbol{k}_1)}{\frac{1}{2}k_1^2 + \frac{1}{2}k_2^2 + \frac{x}{2}k_3^2 + q^2}, \tag{63}$$

with

$$\frac{1}{\Xi_{0,0}} = \frac{1}{\xi} + 4\pi y\int \frac{d^3k_{23}}{(2\pi)^3} \frac{|\chi(k_{23})|^2}{yk_{23}^2 + k_1^2\frac{2x+1}{4y} + q^2}, \tag{64}$$

$$\frac{1}{\Xi_{1,m}} = \frac{1}{g_m} + 4\pi\int \frac{d^3k_{12}}{(2\pi)^3} \frac{|\phi_m(k_{12})|^2 k_{12}^2}{k_{12}^2 + \frac{2x+1}{4}k_3^2 + q^2}, \tag{65}$$

which can be cast in the form of the right-hand sides of Eqs. (11-12) by relabelling the variables $k_{23}$ and $k_{12}$ to $k$ and introducing the relative momenta $\kappa$ given by Eqs. (13-14).

Finally, using Eqs. (57-59), one can make a change of the integration variables in Eq. (62-63) such that they correspond to the variables of the functions $P_m$ and $S$. This change shows that the terms proportional to $S(k_2)$ and $S(k_1)$ in Eq. (63) give the same contribution, which leads to the equations,

$$\frac{S(\boldsymbol{k}_1)}{4\pi\Xi_{0,0}} = y \int \frac{d^3 k_2}{(2\pi)^3} \frac{\langle\chi|\boldsymbol{k}_2 + \frac{x}{2y}\boldsymbol{k}_1\rangle\langle\boldsymbol{k}_1 + \frac{x}{2y}\boldsymbol{k}_2|\chi\rangle S(\boldsymbol{k}_2)}{y\left(k_1^2 + k_2^2\right) + x\boldsymbol{k}_1\cdot\boldsymbol{k}_2 + q^2} \tag{66}$$

$$+ 3 \int \frac{d^3 k_3}{(2\pi)^3} \frac{\langle\chi|\boldsymbol{k}_3 + \frac{\boldsymbol{k}_1}{2y}\rangle \sum_m \langle\boldsymbol{k}_1 + \frac{\boldsymbol{k}_3}{2}|\Phi_m\rangle P_m(\boldsymbol{k}_3)}{k_1^2 + y k_3^2 + \boldsymbol{k}_3\cdot\boldsymbol{k}_1 + q^2},$$

$$\frac{P_m(\boldsymbol{k}_3)}{4\pi\Xi_{1,m}} = 2y \int \frac{d^3 k_1}{(2\pi)^3} \frac{\langle\Phi_m|\boldsymbol{k}_1 + \frac{\boldsymbol{k}_3}{2}\rangle\langle\boldsymbol{k}_3 + \frac{\boldsymbol{k}_1}{2y}|\chi\rangle S(\boldsymbol{k}_1)}{k_1^2 + y k_3^2 + \boldsymbol{k}_1\cdot\boldsymbol{k}_3 + q^2}. \tag{67}$$

Relabelling the non-integrated vector as $\boldsymbol{k}$ and the integrated vector as $\boldsymbol{k}'$ in both equations, one arrives at the STM equations (6-7). The quantities $\Xi_{0,0}$ and $\Xi_{1,m}$ are directly related to the two-body scattering amplitudes of the potentials $\hat{V}$ and $\hat{U}$ given respectively by:

$$\frac{1}{\Xi_{0,0}} = -\frac{|\chi(i\kappa_0)|^2}{f_{0,0}(i\kappa_0)}, \tag{68}$$

$$\frac{1}{\Xi_{1,m}} = \frac{|\kappa\phi_m(i\kappa_1)|^2}{f_{1,m}(i\kappa_1)}, \tag{69}$$

where $\kappa_0$ and $\kappa_1$ are defined in Eqs. (13-14).

## C  Form of the functions $S$ and $P$

Here we derive the forms of $S$ and $\boldsymbol{P}$ functions for a three-body state of total angular momentum and parity symmetry $J^\pi = 1^-$. The same considerations of symmetry were done in Ref. [49] in the case of three identical fermions.

### C.1  Form of $S$

$S$ depends on the momentum $\boldsymbol{k}$ between the fermion-particle subsystem (13) and fermion 2. Since the fermion-particle susbsystem is assumed to have an angular momentum $l = 0$ and the total angular momentum is $J = 1$, the only possible angular momentum $L$ between (13) and 2 is 1. The function $S$ is thus proportional to an angular momentum state $L = 1$, with a proportionality factor that depends on the norm $k$ of $\boldsymbol{k}$.

$$S(\boldsymbol{K}) \propto |\overset{J}{1}, \overset{M_J}{0}\rangle_{0\oplus 1}$$

$$\propto |\overset{l}{0}, \overset{m}{0}\rangle|\overset{L}{1}, \overset{M}{0}\rangle$$

$$= s(k)(\boldsymbol{e}_k \cdot \boldsymbol{e}_z), \tag{70}$$

which is Eq. (22). Here, we use the fact that the angular momentum $|\overset{L}{1}, \overset{M}{0}\rangle$ is proportional to $\boldsymbol{e}_k \cdot \boldsymbol{e}_z$, where $\boldsymbol{e}_k = \boldsymbol{k}/k$ is the unit vector along $\boldsymbol{k}$ and $\boldsymbol{e}_z$ is a fixed unit vector along which the projection of angular momentum is assumed to be zero.

## C.2  Form of $P$

$P$ depends on the momentum $\boldsymbol{k}$ between the fermion-fermion subsystem (12) and third particle 3. Since the fermion-fermion subsystem is assumed to have an angular momentum $l = 1$ and the total angular momentum is $J = 1$, the only possible angular momentum $L$ between (12) and 3 is either 0 or 2. From the negative parity, we further restrict to the two angular momentum compositions:

$$| \overset{l}{1}, \overset{m}{0} \rangle | \overset{L}{0}, \overset{M}{0} \rangle \tag{71}$$

and

$$\sqrt{\frac{3}{10}} | \overset{l}{1}, \overset{m}{-1} \rangle | \overset{L}{2}, \overset{M}{1} \rangle - \sqrt{\frac{2}{5}} | \overset{l}{1}, \overset{m}{0} \rangle | \overset{L}{2}, \overset{M}{0} \rangle + \sqrt{\frac{3}{10}} | \overset{l}{1}, \overset{m}{1} \rangle | \overset{L}{2}, \overset{M}{-1} \rangle. \tag{72}$$

The quantity $\boldsymbol{P}(\boldsymbol{k}) \cdot \boldsymbol{k}_{12}$ is therefore a linear combination of the above two angular momentum states, where the linear coefficients depend on the norm $K$. Expressing these angular momentum states in terms of scalar and vector products, one finds:

$$\boldsymbol{P}(\boldsymbol{k}) = p_0(k)\boldsymbol{e}_z + p_2(k)[\boldsymbol{e}_k \times (\boldsymbol{e}_k \times \boldsymbol{e}_z)], \tag{73}$$

which is Eq. (23).

# D  Perturbation of STM equations

We write the STM equation at negative energy in the generic form

$$\int \frac{d^3 k'}{(2\pi)^3} \langle \boldsymbol{k} | \mathcal{M}(q) | \boldsymbol{k}' \rangle \langle \boldsymbol{k}' | \lambda \rangle = 0, \tag{74}$$

where $\mathcal{M}(q)$ may be a matrix as in Eqs. (6, 7) or a scalar operator, as it is the case for example when there is no p-wave interaction. We divide the STM operator into two terms: $\mathcal{M}(q) = \mathcal{M}^{(0)}(q) + \mathcal{M}^{(1)}(q)$ and consider the situation where $\mathcal{M}^{(1)}(q)$ can be treated as a perturbation with respect to the dominant term $\mathcal{M}^{(0)}(q)$. At the first order of perturbation, we decompose the eigenvector and the binding wavenumber as $|\lambda\rangle = |\lambda^{(0)}\rangle + |\lambda^{(1)}\rangle$ and $q = q^{(0)} + q^{(1)}$, such that at the lowest order, the STM equation is

$$\mathcal{M}^{(0)}(q^{(0)})|\lambda^{(0)}\rangle = 0. \tag{75}$$

At the first order in perturbation, the STM equation is expanded as

$$\mathcal{M}^{(0)}(q^{(0)})|\lambda^{(1)}\rangle + q^{(1)} \frac{\delta \mathcal{M}^{(0)}}{\delta q}\bigg|_{q^{(0)}} |\lambda^{(0)}\rangle + \mathcal{M}^{(1)}(q^{(0)})|\lambda^{(0)}\rangle = 0. \tag{76}$$

Without loss of generality, we assume that $\langle \boldsymbol{k} | \mathcal{M}^{(0)} | \boldsymbol{k}' \rangle$ and $\langle \boldsymbol{k} | \mathcal{M}^{(1)} | \boldsymbol{k}' \rangle$ are written in a symmetric form, so that the perturbed eigenvector verifies $\langle \lambda^{(0)} | \lambda^{(1)} \rangle = 0$. Applying $\langle \lambda^{(0)} |$ on the left of Eq. (76), one obtains

$$\langle \lambda^{(0)} | \frac{\delta \mathcal{M}^{(0)}}{\delta q} | \lambda^{(0)} \rangle q^{(1)} + \langle \lambda^{(0)} | \mathcal{M}^{(1)} | \lambda^{(0)} \rangle = 0 \tag{77}$$

and thus

$$q^{(1)} = -\frac{\langle \lambda^{(0)} | \mathcal{M}^{(1)} | \lambda^{(0)} \rangle}{\langle \lambda^{(0)} | \frac{\delta \mathcal{M}^{(0)}}{\delta q} | \lambda^{(0)} \rangle}. \tag{78}$$

In the particular case where $\mathcal{M}^{(0)}$ is the scalar STM operator in absence of p-wave interaction, we will show that $\langle\lambda^{(0)}|\frac{\delta\mathcal{M}^{(0)}}{\delta q}|\lambda^{(0)}\rangle$ can be expressed in terms of the sweep parameter $\frac{\partial E}{\partial(1/a)}$. For this purpose, we consider a small variation of the scattering length in the unperturbed STM equation $a \to a + \delta a$, that induces a small change of the binding wavenumber $q^{(0)} \to q^{(0)} + \delta q$. At the first order in $\delta(1/a)$, one has

$$\langle\lambda^{(0)}|\frac{\delta\mathcal{M}^{(0)}}{\delta q}|\lambda^{(0)}\rangle\delta q + \langle\lambda^{(0)}|\frac{\delta\mathcal{M}^{(0)}}{\delta(1/a)}|\lambda^{(0)}\rangle\delta(1/a) = 0. \tag{79}$$

The value of the term $\langle\lambda^{(0)}|\frac{\delta\mathcal{M}^{(0)}}{\delta(1/a)}|\lambda^{(0)}\rangle$ depends on a common factor in $\mathcal{M}$, $\mathcal{M}^{(0)}$, and $\mathcal{M}^{(1)}$. We fix this factor such that $\frac{\delta\mathcal{M}^{(0)}}{\delta(1/a)} = -1$ and thus $\langle\lambda^{(0)}|\frac{\delta\mathcal{M}^{(0)}}{\delta(1/a)}|\lambda^{(0)}\rangle = -\langle\lambda^{(0)}|\lambda^{(0)}\rangle$. Then, using this normalisation of the STM operator and injecting Eq. (79) in Eq. (78), one finds

$$E^{(1)} = -\frac{\partial E^{(0)}}{\partial(1/a)}\frac{\langle\lambda^{(0)}|\mathcal{M}^{(1)}|\lambda^{(0)}\rangle}{\langle\lambda^{(0)}|\lambda^{(0)}\rangle}, \tag{80}$$

where $E^{(0)}$ is the energy of the unperturbed system $E^{(0)} = -\hbar^2 q^{(0)\,2}/(2M)$ and $E^{(1)}$ is the shift in energy resulting from the perturbation: $E^{(1)} = -\hbar^2 q^{(0)} q^{(1)}/M$.

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
