# Peer review of "Shallow Trimers of Two Identical Fermions and One Particle in Resonant Regimes"

_SciPost Physics, doi:SciPost Phys. 12, 185 (2022)_

## Round 1 · Referee Report · Anonymous (Referee 2) · 2022-2-19

Strengths

Well written introduction and conclusion. Methods are well documented. Realistic scenarios of relevance to experiment are considered.

Weaknesses

Presentation of figures weakens readability/clarity of analysis.

Report

The authors study the three-body spectrum for two identical fermions interacting with a third particle. Here they consider how the KM scenario may be impacted by the presence of p-wave interactions between the fermions, finding a range of other trimers which appear based on the mass ratios and relative strengths of interaction. Although reasonably well written, the analysis suffers from the way in which the results are presented in the figures, often in 6 and 8 panel figures which consequently act to obscure the important trends and underlying physics. My comments below are aimed at improving the readability of the manuscript, whose conclusions I feel are important enough to warrant publication in some form.

Comments:

1.) Separable potentials are used throughout. Can the authors comment on their range of validity for both s- and p-wave interactions? I.e. presumably the separable approximation of the full potential breaks down particularly away from the resonance limit when the T-matrix can no longer be approximated by a single pole (unitary pole approximation)?

2.) I found it difficult to follow the avoided crossing arguments due to the confusing nature of Fig. 6. Here it’s not clear at all which curves correspond to the key especially the 3*10^(-2) 1st and 2nd state which are indicated by the same linetype in the key. Could the authors please work on clarifying/simplifying this figure so that the arguments of the text can be followed more readily? Additionally, could they point out explicitly when the hybridization occurs?

3.) It is easy to get lost in the large number and type of trimers discussed in the text. There are the KM, p-wave induced, s-wave induced (different from KM?), Efimov, hybrid, ground, excited, universal etc… It would help the reader tremendously if the curves could be labeled with the corresponding name/type of the trimer. Also, could the authors clarify when an s-wave trimer ceases to be a KM trimer? Is it a matter of when the trimer has left the universal regime? Also, when a universal trimer is mentioned it would be helpful if this was always accompanied by the type (KM, Efimov, etc…)

4.) There are many six and eight panel figures in the text. Admittedly, they are not the easiest to read and they give the manuscript a disjointed feel as one inevitably has to flip back and forth many times to digest the discussed trend. I ask the authors if so many values of 1/v necessary to convey the trends? And if so, perhaps they can be plotted more compactly. For instance, Fig. 5 could be reduced to two panels: one plot with the ground + KM limit for all values of 1/v and another plot with the exited + KM limit for all values of 1/v.

5.) One of the attractive features of the KM trimers is that their appearance can be expected to be associated with less inelastic losses than Efimov trimers. Can the authors comment on how they expect the stability of the system to change due to the additional trimers that arise when p-wave interactions are considered—in particular the unexpectedly robust presence of the Borromean trimers.

Minor Comments:

1.) When the “three regimes” are discuss on pp. 6, two regimes fall in the region r_e/a<0.01 however it’s not possible to see this region in the plot. Could the authors put an inset zooming into this region? Additionally, there are many plots of the trimer/dimer energy ratio throughout the text although these are never discussed. Are they all necessary?

2.) “s wave” is missing a hyphen below point iv) on pp. 4.

3.) In the first sentence of the second paragraph of Sec. 3, the phrase “for large enough scattering lengths a” is repeated.

4.) It is possible that the bra above Eq. 45 is missing a “|” in \hat{V}\Psi.
  • validity: top
  • significance: good
  • originality: good
  • clarity: good
  • formatting: good
  • grammar: excellent

Author:  Pascal Naidon  on 2022-03-28  [id 2331]

(in reply to Report 2 on 2022-02-19)

(1) Separable potentials are used throughout. Can the authors comment on their range of validity for both s- and p-wave interactions? I.e. presumably the separable approximation of the full potential breaks down particularly away from the resonance limit when the T-matrix can no longer be approximated by a single pole (unitary pole approximation)?

Indeed, a comment on the validity of the separable potential approximation used in our work was missing. The validity of the separable potential approximation is not necessarily related to the separability of the t-matrix. It is generally valid for relative momenta between the two colliding particles that are small as compared to the inverse range of the potential, given by $\Lambda_0$ for the s wave and $\Lambda_1$ for the p wave. We have added this important remark to the main text (after Eq. 15-16) along with some references.

(2) I found it difficult to follow the avoided crossing arguments due to the confusing nature of Fig. 6. Here it’s not clear at all which curves correspond to the key especially the 3*10^(-2) 1st and 2nd state which are indicated by the same linetype in the key. Could the authors please work on clarifying/simplifying this figure so that the arguments of the text can be followed more readily? Additionally, could they point out explicitly when the hybridization occurs?

Following the referee’s suggestion, we clarified the Fig. 6 by removing the crosses corresponding to the state obtained in the KM theory and using different colours to better show the hybridisation (s-wave induced trimer: blue; p-wave induced trimer: red; hybridised: magenta). We extended the legend accordingly.

(3) It is easy to get lost in the large number and type of trimers discussed in the text. There are the KM, p-wave induced, s-wave induced (different from KM?), Efimov, hybrid, ground, excited, universal etc… It would help the reader tremendously if the curves could be labeled with the corresponding name/type of the trimer. Also, could the authors clarify when an s-wave trimer ceases to be a KM trimer? Is it a matter of when the trimer has left the universal regime? Also, when a universal trimer is mentioned it would be helpful if this was always accompanied by the type (KM, Efimov, etc…)

We thank the referee for pointing out this problem of presentation. We agree that the terminology can be confusing for the reader. We therefore added a summary figure and description in the introduction, explaining simply what we mean by the different terms.

To answer the referee’s question, the s-wave induced trimer indeed ceases to be a KM trimer when it leaves the universal regime given by the KM zero-range theory. Of course, this is a continuous transition, and the threshold is arbitrary.

Following the referee’s suggestion, we have added some labels in the figures, to indicate the type of trimer when it is possible.

(4) There are many six and eight panel figures in the text. Admittedly, they are not the easiest to read and they give the manuscript a disjointed feel as one inevitably has to flip back and forth many times to digest the discussed trend. I ask the authors if so many values of 1/v necessary to convey the trends? And if so, perhaps they can be plotted more compactly. For instance, Fig. 5 could be reduced to two panels: one plot with the ground + KM limit for all values of 1/v and another plot with the exited + KM limit for all values of 1/v.

Concerning Fig. 5, we understand what the referee proposes. It would be indeed more compact, but presented in this way, we are afraid that it would be more confusing for a non-specialist to understand the crossing. For this reason, we prefer to keep all the panels in Fig. 5, as they constitute the key frames of an animation where one can see the crossing discussed in the text. On the other hand, we agree that our manuscript has too many figures. Accordingly, we have made the following reductions: Fig. 7: reduction to three panels: 1/v=-10, -1, and 0. Fig. 8: reduction to four panels: alpha = 0.5, 1, 2, 8. Additionally, as suggested elsewhere by the referee, we have removed the panel showing trimer/dimer energy ratio in Fig. 4, as it is not necessary.

(5) One of the attractive features of the KM trimers is that their appearance can be expected to be associated with less inelastic losses than Efimov trimers. Can the authors comment on how they expect the stability of the system to change due to the additional trimers that arise when p-wave interactions are considered—in particular the unexpectedly robust presence of the Borromean trimers.

That is a very important question. We have started to investigate this point, but we have realised it deserves a thorough study, which would involve too much information that can be gathered in a single paper.

Minor Comments:

(1) When the “three regimes” are discuss on pp. 6, two regimes fall in the region $r_e/a<0.01$ however it’s not possible to see this region in the plot. Could the authors put an inset zooming into this region?

We first tried to comply to the referee’s request by adding an inset zoom, but it appears that it brings very little information to the original figure, while making it more visually complex and hiding some parts of the curves. Essentially, in the region $r_e/a≲0.01$ all the curves are indistinguishable, except the one for the cutoff model, and they are all very close to the dashed line (within 0.6%). In the region $r_e/a≲0.001$, all the curves become indistinguishable from the dashed line, and deviate from the KM limit by at most 0.2%. These features are now quantitatively explained in the main text, and we believe they can be understood from the current figure without additional zoom.

Additionally, there are many plots of the trimer/dimer energy ratio throughout the text although these are never discussed. Are they all necessary?

Indeed, they are not, and we have suppressed these panels in Fig. 4.

(2) “s wave” is missing a hyphen below point iv) on pp. 4.

We thank the referee for carefully reading our manuscript. Here, “s wave” is not a compound that modifies a noun, like in “s-wave amplitude” but a noun itself “the s wave” and therefore does not require a hyphen.

(3) In the first sentence of the second paragraph of Sec. 3, the phrase “for large enough scattering lengths a” is repeated.

Indeed, this is an error on our part. We have removed the first occurrence. We thank the referee for their careful reading.

(4) It is possible that the bra above Eq. 45 is missing a “|” in $\hat{V}\Psi$.

It appears that both notations are possible (see for example, Paul Dirac’s lectures). We do not have any particular preference, and following the referee’s suggestion, we have added a vertical line.

Attachment:

diff_jEYanoA.pdf

---

## Round 1 · Referee Report · Anonymous (Referee 1) · 2022-2-19

Strengths

  1. Discovery of Borromean trimer state of two fermions with p-wave interactions, interacting in s-wave with a third particle

  2. Discovery of existence of such trimers at all mass ratios, making them potentially easier to observe experimentally than previously predicted trimers

Weaknesses

  1. The key equations 6 and 7 are stated rather than derived

  2. The discussion about the perturbative approach to anisotropic p-wave interactions appears rushed

Report

The manuscript by Naidon et al. investigates the three-body problem of two identical fermions and 1 distinct particle. This problem has previously been shown to feature universal 3-body (trimer) states above a certain mass ratio in the presence of an s-wave resonance between each of the fermions and the third particle. The novelty of the present work is to consider additionally a fermion-fermion interaction in the p-wave channel. As the authors show, this new interaction leads to the possibility of having a trimer at all mass ratios, and of having Borromean trimers that exist even in the absence of any two-body bound states. These are both remarkable consequences of such a simple change in the setup.

Overall, I found the manuscript to be both very interesting and very clearly written. It contains a detailed abstract and introduction, the results are clearly laid out, the properties of the trimers are discussed in detail, and the conclusions are appropriate. I did find some of the technical aspects difficult to follow (see below), and the referencing could possibly be improved - for instance I found it odd that the sentence about the experimental observation of Efimov physics did not contain a reference to said experiment. Having said that, the present work definitely presents a groundbreaking theoretical discovery, and I believe it fulfils all the SciPost Physics acceptance criteria. I therefore recommend acceptance to SciPost Physics once the following comments have been addressed.

Requested changes

  1. I would strongly encourage the authors to provide a derivation of the key equations 6 and 7.

  2. Equations 4-18 (or almost half of the paper's equations) appear in a single very hard to read paragraph. I would consider breaking that up and expanding the corresponding discussion, including explaining the meaning of the quantities introduced in eqs 11-14.

  3. Could the authors include a brief discussion of how one would in practice measure (directly or indirectly) these trimer states?

  4. Figure 1: Some of the lines are not visible in the top panel. I would explicitly mention that this is the case, and that they can be distinguished in the bottom panel.

  5. Page 6, paragraph containing eq 31: I do not understand the statement "these models correspond to a force that suppresses the wave function". What is it precisely in the models that correspond to a force?

  6. In the last paragraph of section 4 it is mentioned that there are deep bound states and that these can merge with the excited trimer state. Is this feature meant to be visible in Fig 5? If so, could the authors be explicit about where? If not, why not?

  7. I believe the introduction of anisotropy in the p-wave interaction (section 5) breaks the degeneracy of the trimers. Can the splitting be estimated in general or calculated for a specific example?

  8. In general, I found the discussion of the perturbative approach to be a bit rushed. While all other points in the manuscript were well illustrated, there were no corresponding figures, even though these could provide additional insights for the reader. Also, I found the statement that "Using Eq. (40), one finds a shift in energy which exactly equals zero" to be very confusing, given that the authors then proceed to discuss how to obtain the change in the spectrum. Perhaps I am misunderstanding something?

  9. In eq. 38, I was wondering if P(k) in the first term should have been P_M(k) like in eq 7? And from where did the vector k'+k/2 originate that appears in the numerator?

  • validity: top
  • significance: high
  • originality: high
  • clarity: high
  • formatting: excellent
  • grammar: excellent

Author:  Pascal Naidon  on 2022-03-28  [id 2330]

(in reply to Report 1 on 2022-02-19)

Here are our replies to the first referee.

The referencing could possibly be improved - for instance I found it odd that the sentence about the experimental observation of Efimov physics did not contain a reference to said experiment.

We thank the referee for pointing out this shortcoming. We have now added references to experimental observation of Efimov physics.

(1) I would strongly encourage the authors to provide a derivation of the key equations 6 and 7.

We have added an appendix with the detailed derivation.

(2) Equations 4-18 (or almost half of the paper's equations) appear in a single very hard to read paragraph. I would consider breaking that up and expanding the corresponding discussion, including explaining the meaning of the quantities introduced in eqs 11-14.

We have expanded the discussion of these equations and explained their meaning in more detail.

(3) Could the authors include a brief discussion of how one would in practice measure (directly or indirectly) these trimer states?

We thank the referee for making this important remark. We have now added a section at the end of the manuscript discussing how the trimer states of this study could be observed experimentally.

(4) Figure 1: Some of the lines are not visible in the top panel. I would explicitly mention that this is the case,, and that they can be distinguished in the bottom panel.

We have mentioned this point explicitly in the legend.

(5) Page 6, paragraph containing eq 31: I do not understand the statement "these models correspond to a force that suppresses the wave function". What is it precisely in the models that correspond to a force?

We thank the referee for pointing out this wrong formulation. We had in mind the Lennard-Jones model, but indeed, there is no such force in the Gaussian, Yamaguchi and Yamaguchi-squared models, since they are all purely attractive. We simply meant that the interaction in all these models suppresses the wave function in the range $\bar{r}_e$ while barely affecting it outside. We have now made this precise statement in the text.

(6) In the last paragraph of section 4 it is mentioned that there are deep bound states and that these can merge with the excited trimer state. Is this feature meant to be visible in Fig 5? If so, could the authors be explicit about where? If not, why not?

This feature is not meant to be visible in Fig 5. It is just a point we thought would be worthwhile to mention, but not necessarily illustrated by a dedicated figure, as it is a relatively minor point and model dependent. We understand however that the phrasing gives the impression it can be seen in Fig. 5. Therefore, we clarified in the text that it cannot be seen in the figure.

(7) I believe the introduction of anisotropy in the p-wave interaction (section 5) breaks the degeneracy of the trimers. Can the splitting be estimated in general or calculated for a specific example?

Yes indeed, the degeneracy is broken. The splitting has been estimated in our work for a small anisotropy (see next point). Thanks to the referee’s remark, we realised that our presentation was not explicit enough about this important feature. We have changed the preamble of this section accordingly.

(8) In general, I found the discussion of the perturbative approach to be a bit rushed. While all other points in the manuscript were well illustrated, there were no corresponding figures, even though these could provide additional insights for the reader. Also, I found the statement that "Using Eq. (40), one finds a shift in energy which exactly equals zero" to be very confusing, given that the authors then proceed to discuss how to obtain the change in the spectrum. Perhaps I am misunderstanding something?

In this part, we have shown that for a small anisotropy, i.e. for a small relative change of the scattering amplitude for different values of the magnetic number m, the energy of the trimers for a given m is obtained formally from the isotropic STM equation where one uses the scattering amplitude $f_{1,m}$. The shifts in the spectrum are thus simply obtained in this perturbative treatment. We are grateful to the referee for pointing out the possible misunderstanding in our initial formulation. We have changed the ambiguous sentences to clearer statements.

(9) In eq. 38, I was wondering if P(k) in the first term should have been P_M(k) like in eq 7? And from where did the vector k'+k/2 originate that appears in the numerator?

In the STM equation (7), we use the components $P_m(k)$ and the form factor $\Phi_m(\vec{k})=\phi(k) \vec{k}.\hat{e}_m$ which contains a dependence on the orientation of $\vec{k}$, whereas in Eq. (38) we use the vector notation $\vec{P}(k)$ of Eq. (19) and the reduced from factor $\phi(k)$ which depends only on the norm of $\vec{k}$. This is why the vector $\vec{k}’+\vec{k}/2$ appears in the numerator of Eq. (38). We chose this vector notation in Eq. (38) to highlight the symmetry breaking due to the anisotropy. The referee’s remark made us realise that it would have been easier to understand if we first wrote the STM equations in vector form for the isotropic case. This form can also help one to derive Eqs. (22,23,24). Accordingly, we have now written explicitly the STM equations in vector form in Section 2.2, just after introducing the vector notation $\vec{P}(k)$ of Eq. (19).

The changes to the manuscript are shown in the attached file.

Attachment:

diff.pdf

Author:  Pascal Naidon  on 2022-04-06  [id 2360]

(in reply to Pascal Naidon on 2022-03-28 [id 2330])
Category:
correction

We realised that there are some typos in the new equations (20-21) introduced the revised version of the paper. We are afraid these typos may confuse the referees, in particular the first referee who asked (question 9) about the vector appearing in one of the equations. To avoid any confusion, here are the correct equations:
\[
-\frac{\vert\chi(i\kappa_{0})\vert^{2}}{4\pi f_{0}(i\kappa_{0})}S(\mathbf{k})=\qquad\qquad\qquad\qquad\qquad\qquad\\y\int\frac{d^{3}k^{\prime}}{(2\pi)^{3}}\frac{\chi^{*}(\vert\mathbf{k}^{\prime}+\frac{x}{2y}\mathbf{k}\vert)\chi(\vert\mathbf{k}+\frac{x}{2y}\mathbf{k}^{\prime}\vert)S(\mathbf{k}^{\prime})}{y(k^{2}+k^{\prime2})+x\mathbf{k}\cdot\mathbf{k}^{\prime}+q^{2}}\\+3\int\frac{d^{3}k^{\prime}}{(2\pi)^{3}}\frac{\chi^{*}(\vert\mathbf{k}^{\prime}+\frac{1}{2y}\mathbf{k}\vert)\phi(\vert\mathbf{k}+\frac{1}{2}\mathbf{k}^{\prime}\vert)\left(\mathbf{k}+\frac{1}{2}\mathbf{k}^{\prime}\right)\cdot\mathbf{P}(\mathbf{k}^{\prime})}{k^{2}+yk^{\prime2}+\mathbf{k}\cdot\mathbf{k}^{\prime}+q^{2}}
\]

\[
\frac{\vert\kappa_{1}\phi(i\kappa_{1})\vert^{2}}{4\pi f_{1}(i\kappa_{1})}\mathbf{P}(\mathbf{k})=\\2y\int\frac{d^{3}k^{\prime}}{(2\pi)^{3}}\frac{\left(\mathbf{k}^{\prime}+\frac{\mathbf{k}}{2}\right)\phi^{*}(\vert\mathbf{k}^{\prime}+\frac{\mathbf{k}}{2}\vert)\chi(\vert\mathbf{k}+\frac{\mathbf{k}^{\prime}}{2y}\vert)S(\mathbf{k}^{\prime})}{yk^{2}+k^{\prime2}+\mathbf{k}\cdot\mathbf{k}^{\prime}+q^{2}}
\]

We apologise for these typos which we will correct in the final version.

---

## Round 2 · Referee Report · Anonymous (Referee 2) · 2022-4-3

Report

The comments raised in the first round of refereeing have been answered satisfactorily, and I find the manuscript improved and suitable for recommendation to be published in SciPost Physics.

---

## Round 2 · Referee Report · Anonymous (Referee 1) · 2022-4-13

Report

The authors have satisfactorily answered all of my comments. I particularly applaud the authors' significantly enhanced discussion of how they arrive at their main equations, including a new Appendix, as well as the new section on the potential for experimental observation. I am therefore happy to recommend publication in SciPost Physics.

Requested changes

There is a typo in "Schrödinger" at the bottom of page 17, first column.

---

## Editorial Decision

published